# GABAergic synaptic scaling is triggered by changes in spiking activity rather than AMPA receptor activation

Carlos Gonzalez-Islas[1,2], Zahraa Sabra[3], Ming-fai Fong[1,4], Pernille Yilmam[1], Nicholas Au Yong[3], Kathrin Engisch[5], Peter Wenner[1]*

[1]Department of Cell Biology, Emory University, Atlanta, United States; [2]Doctorado en Ciencias Biológicas Universidad Autónoma de Tlaxcala, Tlax, Mexico; [3]Department of Neurosurgery, Emory University, Atlanta, United States; [4]Department of Biomedical Engineering, Georgia Tech and Emory University, Atlanta, United States; [5]Department of Neuroscience, Cell Biology and Physiology, Wright State University, Dayton, United States

*For correspondence:
pwenner@emory.edu

Competing interest: The authors declare that no competing interests exist.

**Abstract** Homeostatic plasticity represents a set of mechanisms that are thought to recover some aspect of neural function. One such mechanism called AMPAergic scaling was thought to be a likely candidate to homeostatically control spiking activity. However, recent findings have forced us to reconsider this idea as several studies suggest AMPAergic scaling is not directly triggered by changes in spiking. Moreover, studies examining homeostatic perturbations *in vivo* have suggested that GABAergic synapses may be more critical in terms of spiking homeostasis. Here, we show results that GABAergic scaling can act to homeostatically control spiking levels. We found that perturbations which increased or decreased spiking in cortical cultures triggered multiplicative GABAergic upscaling and downscaling, respectively. In contrast, we found that changes in AMPA receptor (AMPAR) or GABAR transmission only influence GABAergic scaling through their indirect effect on spiking. We propose that GABAergic scaling represents a stronger candidate for spike rate homeostat than AMPAergic scaling.

## eLife assessment

This is an **important** study that brings insight into mechanisms that underlie regulation of GABAergic transmission in response to changes in activity. The authors present **solid** data supporting the premise that action potential firing rather than excitatory synaptic strength is a key determinant of GABAergic synaptic inputs.

## Introduction

Homeostatic plasticity represents a set of compensatory mechanisms that are thought to be engaged by the nervous system in response to cellular or network perturbations, particularly in developing systems (*Tien and Kerschensteiner, 2018*). Synaptic scaling is one such mechanism where homeostatic compensations in the strength of the synapses onto a neuron occur following chronic perturbations in spiking activity or neurotransmitter receptor activation (neurotransmission) (*Turrigiano et al., 1998*). Scaling is typically identified by comparing the distribution of miniature postsynaptic current (mPSC) amplitudes in control and activity-perturbed conditions. For instance, when spiking activity in cortical cultures was reduced for 2 days with the Na$^+$ channel blocker TTX or the AMPA/kainate glutamate receptor antagonist CNQX, mEPSC amplitudes were increased (*Turrigiano et al., 1998*). When first

discovered, homeostatic synaptic scaling was thought to be triggered by the cell sensing its reduction in spike rate through reduced calcium entry into the cytoplasm. This was then believed to alter global calcium signaling cascades that led to increased AMPA receptor (AMPAR) insertion in a cell-wide manner such that all synapses increased synaptic strength multiplicatively based on each synapse's initial strength (*Turrigiano, 2012*). In this way excitatory synaptic strength was increased across all of the cell's inputs in order to recover spiking activity without altering relative synaptic strengths resulting from Hebbian plasticity mechanisms. These criteria, sensing spike rate and adjusting synaptic strengths multiplicatively, thus established the expectations for homeostatic synaptic scaling and were consistent with the idea that AMPAergic scaling could be a spike rate homeostat.

More recent work has demonstrated that AMPAergic synaptic scaling is more complicated than originally thought. First, studies have now shown that increases in mEPSC amplitudes or synaptic glutamate receptors often do not follow a simple multiplicative function (*Hanes et al., 2020*; *Wang et al., 2019*). Rather, these studies show that changes in synaptic strength at different synapses exhibit different scaling factors, arguing against a single multiplicative scaling factor that alters synaptic strength globally across the cell. Second, AMPAergic scaling triggered by receptor blockade can induce a synapse-specific plasticity rather than a cell-wide plasticity. Compensatory changes in synaptic strength were observed in several studies where neurotransmission at individual synapses was reduced (*Hou et al., 2008*; *Sutton et al., 2006*; *Béïque et al., 2011*; *Deeg and Aizenman, 2011*). This synapse-specific plasticity would appear to be cell-wide if neurotransmission at all synapses were reduced as occurs in the typical pharmacological blockades that are used to trigger scaling. Regardless, this would still be a synapse specific plasticity, determined at the synapse, rather than the cell sensing its lowered spiking activity through global calcium levels. Finally, several different studies now suggest that reducing spiking levels in neurons is not sufficient to trigger AMPAergic upscaling and therefore bring into question its role as a spike rate homeostat. Forced expression of a hyperpolarizing conductance reduced spiking of individual cells but did not trigger AMPAergic scaling (*Burrone et al., 2002*). Further, optogenetic restoration of culture-wide spiking in the presence of AMPAergic transmission blockade triggered AMPAergic scaling that was indistinguishable from that of cultures where AMPAR block reduced spiking (no optogenetic restoration of spiking) (*Fong et al., 2015*). Most studies that separate the importance of cellular spiking from synapse-specific transmission suggest that AMPAergic scaling is triggered by changes in neurotransmission, rather than a cell's spiking activity (*Deeg and Aizenman, 2011*; *Burrone et al., 2002*; *Fong et al., 2015*; *Garcia-Bereguiain et al., 2016*). While transmission-dependent AMPAergic scaling appears to be more commonly observed, there are two studies that suggest that alterations in AMPAergic synaptic strength can occur following alterations in spiking in individual cells - AMPAR accumulation following blockade of spiking at the soma in cortical cultures (*Ibata et al., 2008*) and reduced mEPSC amplitude following optogenetic activation of individual cells in hippocampal cultures (*Goold and Nicoll, 2010*).

Because the pharmacological perturbations that trigger AMPAergic upscaling also result in GABAergic downscaling, it is assumed that they have common triggers. Therefore, in the current study we tested this possibility. Homeostatic regulation of GABAergic miniature inhibitory postsynaptic current (mIPSC) amplitude was first shown in excitatory neurons following network activity perturbations (*Kilman et al., 2002*). Similar to AMPAergic upscaling, chronic perturbations in AMPAR or spiking activity triggered mIPSC downscaling through compensatory changes in the number of synaptic $GABA_A$ receptors (*Kilman et al., 2002*; *Swanwick et al., 2006*; *Hartman et al., 2006*; *Peng et al., 2010*; *Wenner, 2011*). However, the sensing machinery for triggering GABAergic scaling may be distinct from that of AMPAergic scaling (*Joseph and Turrigiano, 2017*). Further, GABAergic plasticity does appear to be a key player in the homeostatic response *in vivo*, as many different studies have shown strong GABAergic compensations following somatosensory, visual, and auditory deprivations (*Gainey et al., 2018*; *Li et al., 2014*; *Hengen et al., 2013*; *Barnes et al., 2015*; *Kuhlman et al., 2013*). In addition, these homeostatic GABAergic responses precede and can outlast compensatory changes in the glutamatergic system. Here, we describe that GABAergic scaling is triggered by changes in spiking levels rather than changes in AMPAergic or GABAergic neurotransmission, that GABAergic scaling is expressed in a multiplicative manner, and could contribute to the homeostatic recovery of spiking activity. Our results suggest that GABAergic scaling could serve as a homeostat for spiking activity.

## Results

### TTX and AMPAR blockade triggered a non-uniform scaling of AMPA mPSCs

Previously we have shown that blocking spike activity in neuronal cultures triggered AMPAergic scaling in a non-uniform or divergent manner, such that different synapses scaled with different scaling ratios (*Hanes et al., 2020*; *Koesters et al., 2024*). Importantly, these results were consistent across independent studies performed in three different labs using rat or mouse cortical cultures, or mouse hippocampal cultures. We quantitatively evaluated scaling in the following manner. We rank-ordered mEPSC amplitudes (smallest to largest) for both control and TTX-treated cultures and then divided the TTX rank-ordered amplitude by the corresponding control rank-ordered amplitude (e.g. smallest TTX amplitude divided by smallest control amplitude, etc.) and plotted these ratios for all such comparisons (*Hanes et al., 2020*; *Koesters et al., 2024*). Previously, scaling had been thought to be multiplicative, meaning all mPSC amplitudes were altered by a single multiplicative factor. If true for AMPAergic scaling, then our ratio plots should have produced a horizontal line at the scaling ratio. However, we found that ratios progressively increased across at least 75% of the distribution of amplitude ratios. Still, it was unclear whether this was true for all forms of AMPAergic scaling triggered by different forms of activity blockade. Therefore, we repeated this analysis on the data from our previous study (*Fong et al., 2015*), but now on AMPAergic scaling produced by blocking AMPAR neurotransmission (40 µM CNQX), rather than TTX. We found that the scaling was non-uniform and replicated the scaling triggered by TTX application (*Figure 2—figure supplement 1*). There was an abrupt increase in the ratio from 1 to ~1.2 (steeper slope) over the first 1–2% of the data, consistent with an error caused by the detection threshold (as shown in simulations of a threshold issue in *Hanes et al., 2020*). However, ratios then increased over the vast majority of the data from 1.2 to 1.5 more slowly, and this represented the magnitude of homeostatic plasticity with increasing mEPSC amplitude. The results suggest that AMPAergic scaling produced by blocking glutamatergic transmission or spiking in culture was not multiplicative, but rather different synapses increased by different scaling factors. Further, the similarity of scaling ratio plots following either action potential or AMPAergic blockade is consistent with the idea that they are mediated by similar mechanisms.

### TTX and AMPAR blockade reduced both spiking and GABAergic mIPSC amplitude

Previously we made the surprising discovery that AMPAergic upscaling in rat cortical cultures was triggered by a reduction in AMPAR activation rather than a reduction in spiking activity (*Fong et al., 2015*). Here, we tested whether GABAergic scaling was dependent on AMPAR activation or a different trigger, by changes in spiking activity levels. We plated E18 mouse cortical neurons on 64-channel planar multi-electrode arrays (MEAs) and allowed the networks to develop for ~14 days *in vitro* (DIV), a time point where most cultures develop a network bursting behavior (*Figure 1A*, *Figure 1—figure supplement 1*; *Wagenaar et al., 2006*). We used a custom-written MATLAB program that was able to detect and compute overall spike rate and burst frequency (*Figure 1—figure supplement 1*, see Materials and methods). We again found that TTX abolished bursts and spiking activity (n=2, *Figure 1—figure supplement 2*). On the other hand, AMPAR blockade (20 µM) merely reduced bursts and spiking, with a greater effect on bursting. Similar to our findings in rat cortical cultures (*Fong et al., 2015*), CNQX dramatically reduced burst frequency and maintained this reduction for the entire 24 hr of treatment (*Figure 1B*). Overall spike frequency was also reduced in the first 6 hr, but then recovered over the 24 hr drug treatment (*Figure 1C*). While overall spiking was recovered, we did note that this was highly variable, with some cultures recovering minimally. Following AMPAergic blockade bursts continued in these cultures, likely due to NMDAergic neurotransmission as shown previously (*Fong et al., 2015*).

In order to examine the possibility that compensatory changes in GABAergic synaptic strength could have contributed to the recovery of the network spiking activity, we assessed synaptic scaling by measuring mIPSC amplitudes in pyramidal-like neurons in a separate set of cortical cultures plated on coverslips. We found that both activity blockade with TTX and AMPAergic blockade with CNQX triggered a dramatic compensatory reduction in mIPSC amplitude compared to control (untreated) cultures (*Figure 2A*). Even though TTX completely abolished spiking, while CNQX only reduced

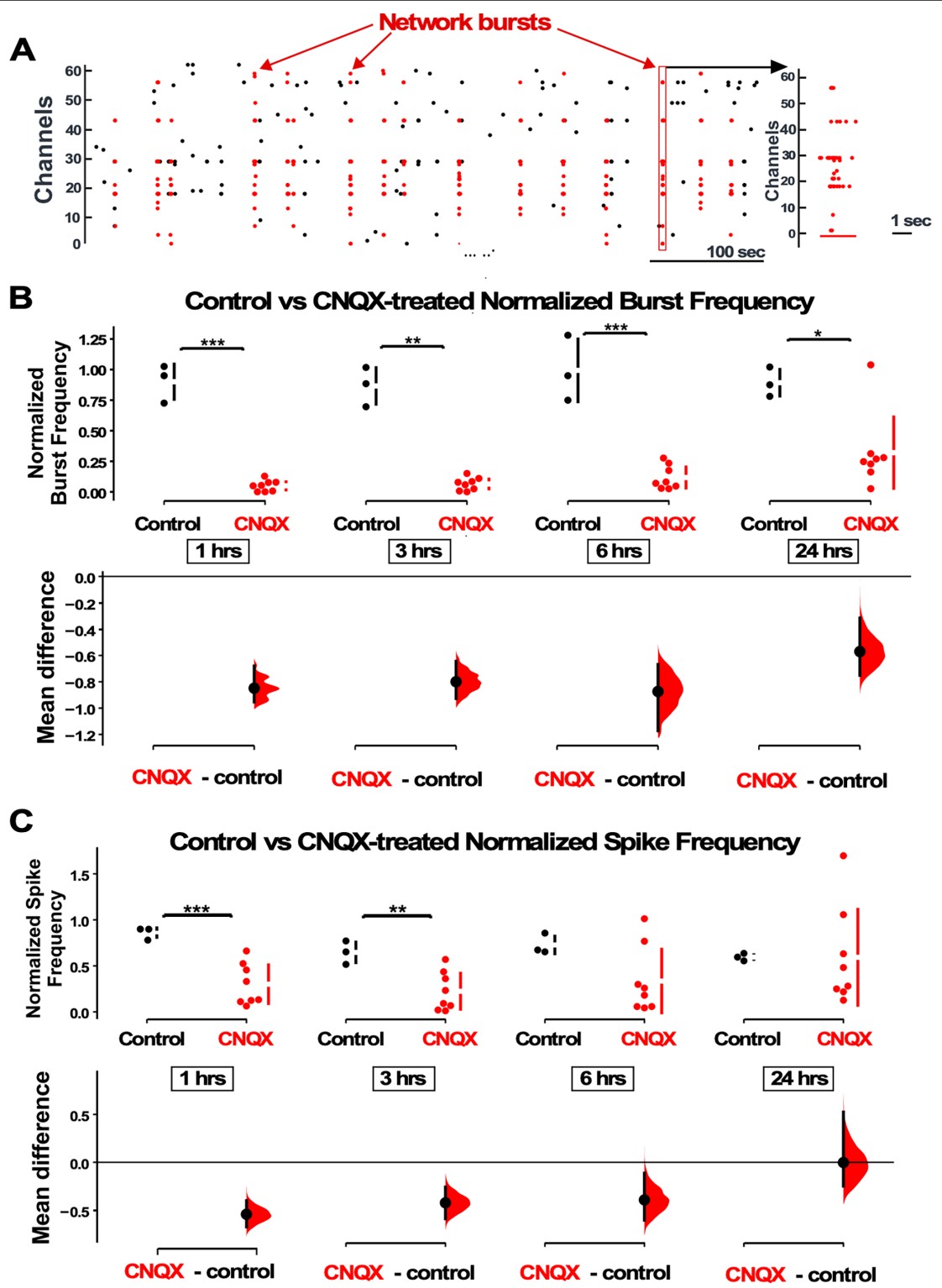

**Figure 1.** AMPAergic blockade reduces burst frequency and overall spike rate. (**A**) Network bursts can be identified by detected spikes (red dots) time-locked in multiple channels of the multi-electrode array (MEA) (Y axis). One burst (red rectangle) is expanded in time and shown in the raster plot on the right. (**B**) The normalized burst rate is shown in control cultures and following application of CNQX for 24 hr. (**C**) Average overall spike frequency is compared for CNQX-treated cultures and control unstimulated cultures at 1 hr, 3 hr, 6 hr (p=0.104), and 24 hr (p=0.982) after addition of CNQX or vehicle. The mean differences at different time points are compared to control and displayed in Cumming estimation plots. Significant differences denoted by *p≤0.05, **p≤0.01, ***p≤0.001. Recordings from single cultures (filled circles, control n=3 cultures, CNQX n=8 cultures), where mean values

*Figure 1 continued on next page*

*Figure 1 continued*

(represented by the gap in the vertical bar) and SD (vertical bars) are plotted on the upper panels. Mean differences between control and treated groups are plotted on the bottom panel, as a bootstrap sampling distribution (mean difference is represented by a filled circles and the 95% CIs are depicted by vertical error bars).

The online version of this article includes the following figure supplement(s) for figure 1:

**Figure supplement 1.** Custom-written MATLAB program identifies bursts in cortical cultures plated on multi-electrode arrays (MEAs) by choosing the minimum number of spikes per burst (Spikes/Burst) across a minimum number of channels contributing to a burst (Min channels) within a maximum Time Window.

**Figure supplement 2.** Rasterplot of cortical culture plated on multi-electrode array (MEA) demonstrating network bursting (red dots, upper plot).

spiking, both treatments triggered a similar reduction in average mIPSC amplitude. In order to more carefully compare the GABAergic scaling that is triggered by TTX and CNQX mechanistically, we created scaling ratio plots as described above (*Hanes et al., 2020*). In *Figure 2B* we show that TTX-induced and CNQX-induced scaling does produce a largely multiplicative downscaling with a scaling factor around 0.5. This is consistent with the idea that the mechanisms of GABAergic scaling were similar following activity or AMPAergic blockade. We noticed that the first mIPSC ratios started near 1 and within 50 ratios came down to the 0.5 level (*Figure 2B*). This is likely due to the smallest drug-treated mIPSCs falling below our detection cutoff of 5 pAs (*Hanes et al., 2020*). On the other hand, the largest mIPSCs trended above 0.5, consistent with the possibility that a small portion of the mIPSCs may not scale uniformly. Together, these results are consistent with the idea that either spiking or reduced AMPAR activation could trigger the GABAergic downscaling since TTX and CNQX both reduce spiking and AMPAR activation.

## Optogenetic restoration of spiking in the presence of AMPAR blockade prevented GABAergic downscaling

In order to separate the importance of spiking levels from AMPAR activation in triggering GABAergic downscaling, we blocked AMPARs while restoring spike frequency. Cultures were plated on the MEA and infected with ChR2 under the human synapsin promoter on DIV 1. Experiments were carried out on ~DIV 14, when cultures typically express network bursting. Baseline levels of spike frequency were measured in a 3 hr period before the addition of 20 μM CNQX (*Figure 3A*). We then used a custom-written TDT Synapse software that triggered a brief (50–100 ms) activation of a blue light photodiode to initiate bursts (see Materials and methods, *Figure 3B*) whenever the running average of the firing rate fell below the baseline level, established before the addition of the drug. In this way we could optically initiate bursts that largely occurred after the blue light was off. These optically induced bursts look very similar to the spontaneously occurring pre-drug bursts and this largely restored the spike rate to pre-drug values (*Figure 3B*). We used 20 μM CNQX to block AMPARs, instead of the 40 μM concentration that we used in the previous study (*Fong et al., 2015*) because 40 μM CNQX severely impaired our ability to optogenetically restore spiking activity in these cultures.

We have already established that bursts and spiking were reduced following the application of CNQX (*Figure 1*). However, when we optogenetically activated the cultures in the presence of CNQX, we found that both the burst rate and spike frequency were increased compared to CNQX treatment alone, no optostimulation (*Figure 3—figure supplement 1*). Because the program was designed to maintain total spike frequency, photostimulation of CNQX-treated cultures did a relatively good job at recovering this parameter to control levels (*Figure 3D*). In fact, spike frequency was slightly, but not significantly, above control levels through the 24 hr recording period (*Figure 3D*). In our previous study we were able to establish that the optogenetically evoked bursts in CNQX and even the pattern of individual unit spiking during the burst was restored to that of normally occurring bursts in the pre-drug condition (*Fong et al., 2015*). On the other hand, the program designed to control the overall spike frequency through optostimulation in CNQX did not completely return burst frequency back to control levels (*Figure 3C*).

We next assessed mIPSC amplitudes using whole-cell recordings taken from cultures plated on MEAs. After blocking AMPAR activation without optogenetic restoration of spiking activity, we found that mIPSC amplitudes were significantly reduced compared to control conditions (*Figure 4A*), as we had shown for CNQX treatment on cultures plated on coverslips (*Figure 2A*). Strikingly, when spiking

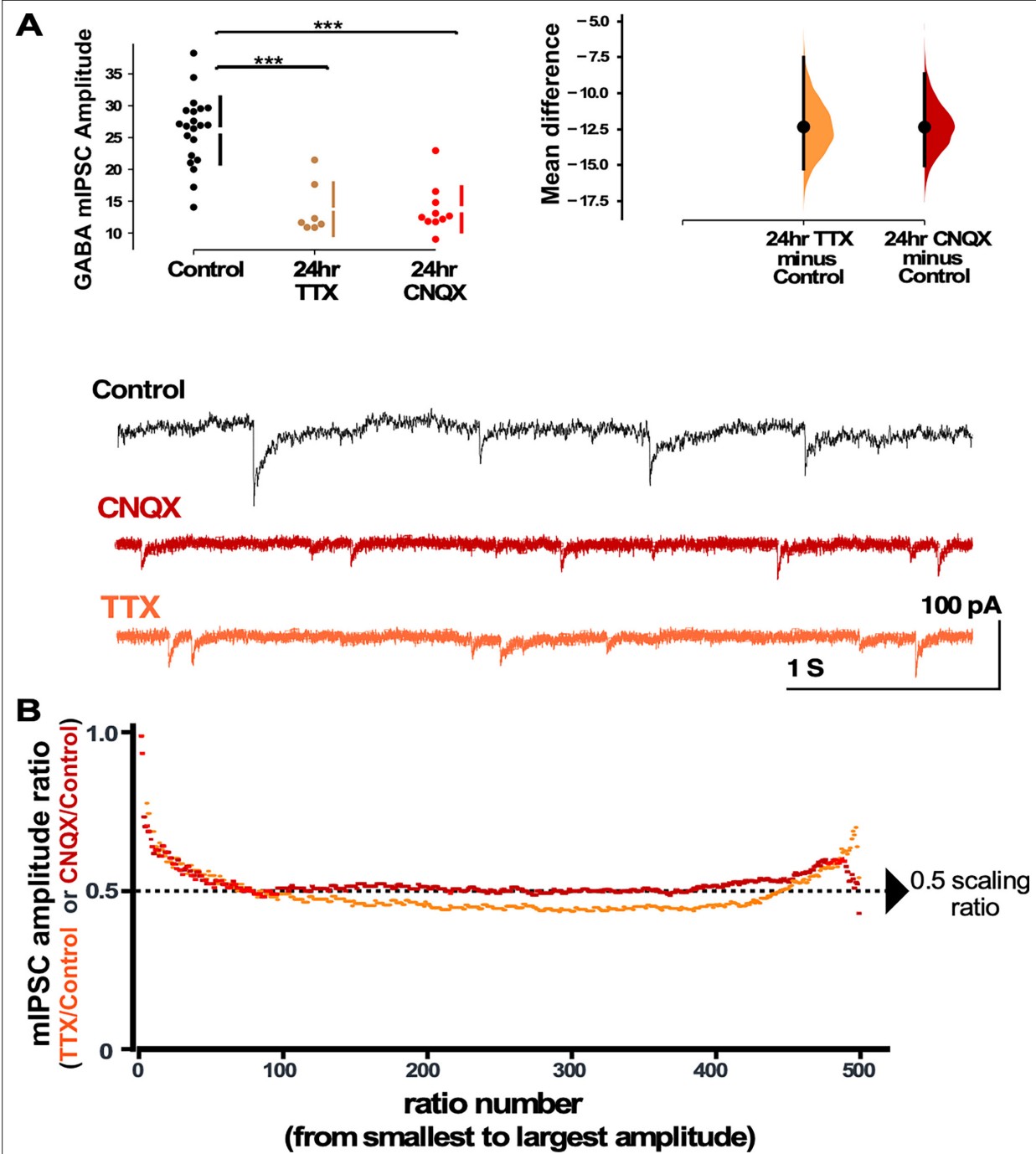

**Figure 2.** Both activity and AMPA receptor (AMPAR) blockade cause a reduction in miniature inhibitory postsynaptic current (mIPSC) amplitudes that appear to scale down. (**A**) CNQX and TTX produce a reduction in average amplitude of mIPSCs as shown in the scatter plots (control - n=21 from 10 cultures, TTX - n=7 from 3 cultures, CNQX - n=10 from 6 cultures). The mean differences are compared to control and displayed in Cumming estimation plots. Significant differences denoted by ***p≤0.001. GABAergic mIPSC amplitudes from single neurons (filled circles), where mean values (represented by the gap in the vertical bar) and SD (vertical bars) are plotted on the panels to the left. Mean differences between control and treated groups are plotted on the panel to the right, as a bootstrap sampling distribution (mean difference is represented by a filled circles and the 95% CIs are depicted by vertical error bars). Example traces showing mIPSCs are shown below. (**B**) Scaling ratio plots show the relationship of mIPSC amplitudes from treated cultures compared to untreated cultures. All recordings taken from cultured neurons plated on coverslips, not multi-electrode arrays (MEAs).

The online version of this article includes the following figure supplement(s) for figure 2:

**Figure supplement 1.** AMPA receptor (AMPAR) block triggered non-uniform AMPAergic scaling.

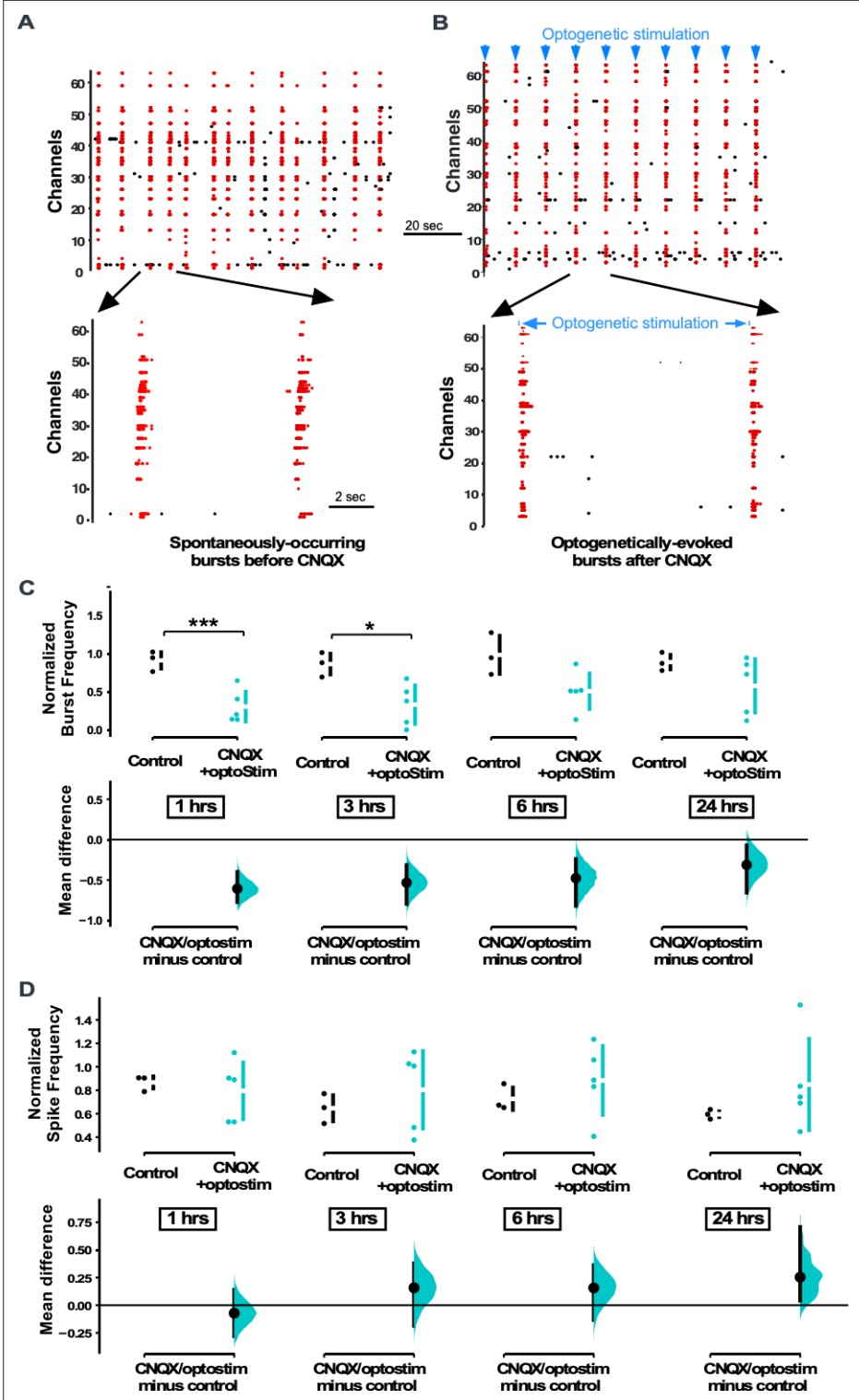

**Figure 3.** Multi-electrode array (MEA) recordings show that optogenetic stimulation restores spiking activity in cultures treated with CNQX. (**A**) Spontaneously occurring bursts of spiking are identified (synchronous spikes/ red dots). Expanded version of raster plot highlighting two bursts is shown below. (**B**) Same as in A, but after CNQX was added to the bath and bursts were now triggered by optogenetic stimulation (blue line shows duration of optogenetic stimulation). (**C**) Average burst rate is compared for CNQX-treated cultures with optogenetic stimulation (n=5 cultures) and control unstimulated cultures (n=3 cultures) at 1 hr, 3 hr, 6 hr (p=0.056), and 24 hr (p=0.379) after addition of CNQX or vehicle (same control data presented in *Figure 1*). (**D**) Average overall spike

*Figure 3 continued on next page*

*Figure 3 continued*

frequency is compared for CNQX-treated cultures with optogenetic stimulation and control unstimulated cultures at 1 hr (p=0.612), 3 hr (p=0.489), 6 hr (p=0.449), and 24 hr (p=0.22) after addition of CNQX or vehicle. Control data is same as presented in *Figure 1*. The mean differences at different time points are compared to control and displayed in Cumming estimation plots. Significant differences denoted by *p≤0.05, ***p≤0.001. Recordings from single cultures (filled circles), where mean values (represented by the gap in the vertical bar) and SD (vertical bars) are plotted on the upper panels. Mean differences between control and treated groups are plotted on the bottom panel, as a bootstrap sampling distribution (mean difference is represented by a filled circles and the 95% CIs are depicted by vertical error bars).

The online version of this article includes the following figure supplement(s) for figure 3:

**Figure supplement 1.** Multi-electrode array (MEA) recordings show optostim + CNQX increases burst frequency and spike frequency compared to CNQX alone.

activity was optogenetically restored in the presence of CNQX for 24 hr, we observed that mIPSCs were no different than control values (same as control, larger than CNQX only - *Figure 4A*). This result suggested that unlike AMPAergic upscaling, GABAergic downscaling was prevented if spiking activity levels were restored in the presence of AMPAR blockade. In order to compare scaling profiles, we plotted the scaling ratios for these different treatments. Not surprisingly, we found that MEA-plated cultures treated with CNQX but given no optogenetic stimulation were similar to CNQX-treated cultures plated on coverslips (CNQX/control ~0.5, *Figure 4B* vs *Figure 2B*). Ratio plots of cultures treated with CNQX where activity was restored optogenetically compared to controls demonstrated a fairly uniform relationship with a ratio of around 1 through most of the distribution suggesting the mIPSCs in these two conditions were similar and therefore unscaled (*Figure 4B*). Interestingly, the scaling ratio and the average mIPSC amplitudes in the optogenetically activated cultures were slightly larger than control mIPSCs which may be due to the slight increase in spiking in optogenetically stimulated cultures. Together, these results suggest that GABAergic downscaling was triggered by reductions in spiking activity, independent of AMPAR activation, and was multiplicative since the vast majority of mEPSC amplitudes (~95%) appeared to be reduced to ~50%.

## Enhancement of AMPAR currents triggered GABAergic upscaling in a spike-dependent manner

While reductions in spiking activity triggered a GABAergic downscaling, it was less clear whether increases in spiking activity could trigger compensatory GABAergic upscaling. To test for such a possibility, we exposed the cultures to cyclothiazide (CTZ), an allosteric enhancer of AMPARs that also enhances spontaneous glutamate release (*Fong et al., 2015*). Due to the hydrophobic nature of CTZ it was necessary to dissolve it in ethanol, and used ethanol without CTZ as a control (final solution 1:1000 ethanol in Neurobasal). In addition to increasing AMPAR activation, CTZ application slightly increased overall spiking activity in our MEA-plated cultures in the first 3 hr of the drug, although this was quite variable (*Figure 5A and B*). The amplitude of mIPSCs in control cultures exposed to ethanol were no different than control cultures without ethanol (*Figure 5C*). We then treated coverslip-plated cultures with CTZ for 24 hr and found that this did indeed produce a compensatory increase in GABA mIPSC amplitude (*Figure 5D*). In our previous study we found that enhancing AMPAergic neurotransmission in the presence of activity blockade (CTZ + TTX) reduced AMPAergic upscaling compared to activity blockade alone (TTX) (*Fong et al., 2015*). Therefore, we tested whether enhancing AMPAergic neurotransmission in activity blockade (CTZ + TTX) altered GABAergic scaling induced by TTX alone (24 hr). Here, we found that the GABAergic downscaling following TTX was no different when AMPAergic neurotransmission was enhanced (CTZ + TTX, *Figure 5D*). To determine if these changes in mIPSC amplitude were of a multiplicative scaling nature, we made ratio plots. This demonstrated that both CTZ increases and CTZ + TTX decreases in mIPSC amplitude were multiplicative and therefore represented scaling (*Figure 5E*, CTZ - scaling ratio of 1.5, CTZ + TTX - scaling ratio of 0.6). Further, the scaling ratio plot for CTZ + TTX looked similar to those of TTX alone (compare *Figures 5E and 2B*). These results showed a compensatory upward and downward GABAergic scaling, that more closely followed spiking activity compared to AMPAergic transmission.

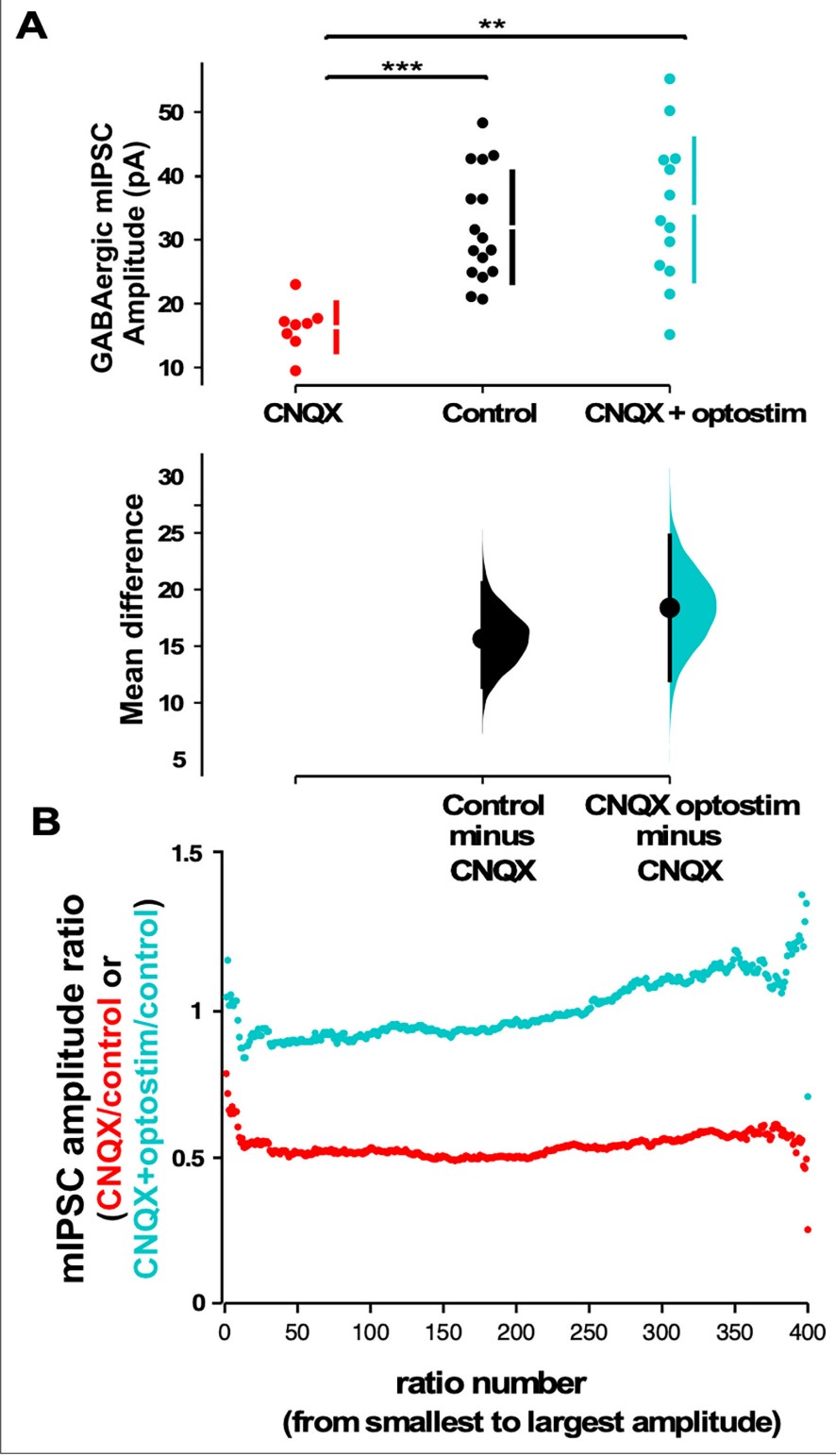

**Figure 4.** Optogenetic restoration of spiking activity in the presence of AMPA receptor (AMPAR) blockade prevents GABAergic downscaling observed in CNQX alone. (**A**) Scatter plots show AMPAR blockade triggers a reduction in miniature inhibitory postsynaptic current (mIPSC) amplitude compared to controls that is prevented when combined with optogenetic stimulation (optostim, control - n=16 from 10 cultures, CNQX - n=8 from 4 cultures, CNQX/optostim - n=13 from 6 cultures). The mean differences are compared to control and displayed in Cumming estimation plots. Significant differences denoted by **p≤0.01, ***p≤0.001. GABAergic mIPSC amplitudes

*Figure 4 continued on next page*

*Figure 4 continued*

from single neurons (filled circles), where mean values (represented by the gap in the vertical bar) and SD (vertical bars) are plotted on the upper panels. Mean differences between control and treated groups are plotted on the bottom panel, as a bootstrap sampling distribution (mean difference is represented by a filled circles and the 95% CIs are depicted by vertical error bars). (**B**) Scaling ratio plots show largely multiplicative relationships to control values for both CNQX and CNQX + optostimulation treatments. Cultured neurons for these recordings were obtained from cells plated on multi-electrode arrays (MEAs) (control, CNQX, and CNQX + optostim).

## Blocking GABAergic receptors for 24 hrs triggered upscaling of GABAergic mIPSCs

The above results suggested that GABAergic scaling was more dependent on the levels of spiking activity. However, one alternative possibility was that these changes in GABA mPSCs were capable of following spike rate changes by using GABAergic receptor activation as a proxy for activity levels (e.g. increased activity is sensed through increased GABAergic receptor activation that then triggers

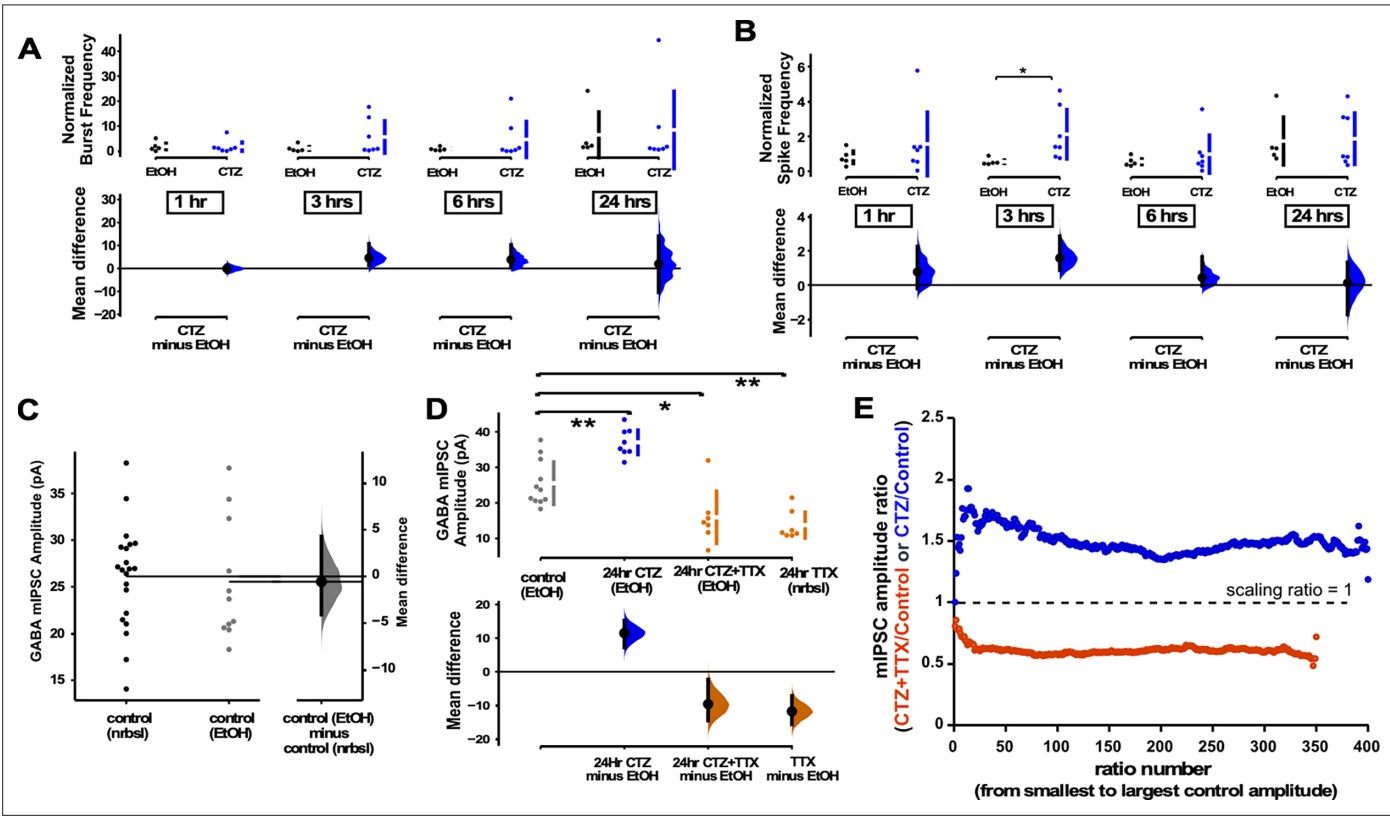

**Figure 5.** GABAergic upscaling was triggered by cyclothiazide (CTZ) and this was dependent on spiking activity. (**A**) Multi-electrode array (MEA) recordings show that CTZ-treated cultures trended toward increases in normalized burst rate compared to control untreated cultures at 1 hr (p=0.97), 3 hr (p=0.246), 6 hr (p=0.397), and 24 hr (p=0.894) after addition of CNQX (n=7) or vehicle (n=5). (**B**) MEA recordings show that CTZ-treated cultures trended toward increases in normalized overall spike rate compared to control untreated cultures at 1 hr (p=0.565), 3 hr, 6 hr (p=0.634), and 24 hr (p=0.92) after addition of CNQX or vehicle. (**C**) Control cultures in Neurobasal (nrbsl) were compared with control cultures with ethanol (EtOH) dissolved in Neurobasal (1:1000). Amplitude of miniature inhibitory postsynaptic currents (mIPSCs) in different controls were no different (p=0.803, nrbsl - n=21 from 10 cultures, EtOH - n=11 from 3 cultures). (**D**) CTZ treatment (dissolved in ethanol) led to an increase in mIPSC amplitude compared to ethanol control cultures (CTZ - n=8 from 3 cultures). CTZ combined with TTX (in ethanol) produced a reduction of mIPSC amplitude compared to controls (ethanol) that was no different than TTX (nrbsl) alone (CTZ + TTX - n=7 from 3 cultures, TTX - n=7 from 3 cultures is same data as shown in *Figure 2A*). The mean differences at different time points or conditions are compared to control and displayed in Cumming estimation plots. Significant differences denoted by *p≤0.05, **p≤0.01. Recordings from single cultures (filled circles), where mean values (represented by the gap in the vertical bar) and SD (vertical bars) are plotted on the upper panels. Mean differences between control and treated groups are plotted on the bottom panel, as a bootstrap sampling distribution (mean difference is represented by a filled circles and the 95% CIs are depicted by vertical error bars). (**E**) Scaling ratios show that both CTZ-induced increases and CTZ + TTX-induced decreases were multiplicative. All mIPSC amplitudes recorded from cultures plated on coverslips, not MEAs.

GABAergic upscaling). In this way, GABARs sense changes in spiking activity levels and directly trigger GABAergic scaling to recover activity. To address this possibility, we treated cultures with the GABA$_A$ receptor antagonist bicuculline to chronically block GABAergic receptor activation while increasing spiking activity. If increased spiking activity is directly the trigger (not mediated through GABAR activity), then we would expect to see GABAergic upscaling. On the other hand, if GABAR activation is a proxy for spiking then blockade of these receptors would indicate low activity levels and we would expect a downscaling to recover the apparent loss of spiking. GABAR block produced an upward trend in both burst frequency (*Figure 6A*) and spike frequency (*Figure 6B*). We measured mIPSCs in a separate cohort of cultures plated on coverslips which were treated with bicuculline for 24 hr, and we observed GABAergic upscaling (*Figure 6C*). These results are consistent with previous work in hippocampal cultures that showed GABAergic upscaling following bicuculline treatment (*Peng et al., 2010*; *Pribiag et al., 2014*). We also assessed mIPSC frequency in all of the drug conditions but did not observe significant differences, possibly due to the tremendous variability of this feature (*Figure 6—figure supplement 1*). Our results were consistent with the idea that direct changes in spiking activity, rather than AMPA or GABA receptor activation, triggered compensatory GABAergic upscaling. The scaling ratio plots were again relatively flat, with a scaling ratio of around 1.5, suggesting a multiplicative GABAergic upscaling (*Figure 6D*) that was similar to CTZ-induced upward scaling (*Figure 5E*).

## The trigger for GABAergic and AMPAergic scaling is distinct in mouse cortical cultures

We have shown the importance of alterations of spiking activity in triggering GABAergic scaling in mouse cortical cultures. Previously, we had shown that AMPAergic scaling was dependent on glutamatergic transmission rather than spiking, and did this in rat cortical cultures. This is a striking result as we had expected these homeostatic mechanisms to share a common trigger. To ensure that the triggers for AMPAergic and GABAergic scaling really were distinct in the same culture set and conditions used in the present study, we repeated our experiment blocking AMPAR activation for 24 hr with 20 µM CNQX, but now checked for AMPAergic scaling. We found the surprising result that following 24 hr CNQX treatment there was no change in AMPAergic mEPSC amplitudes (*Figure 7*), despite the fact that this was the same treatment that reduced spiking activity in our cultures and triggered GABAergic downscaling. The result confirms the observation that the triggers for AMPAergic and GABAergic scaling in the same cultures were distinct.

## Discussion

In the original study describing AMPAergic synaptic scaling, the authors triggered this plasticity by blocking spiking activity with TTX or blocking AMPAergic neurotransmission with CNQX (*Turrigiano et al., 1998*). Similar results have now been demonstrated in multiple tissues and labs (*Koesters et al., 2024*). It was thought that AMPAergic scaling was a homeostatic mechanism, triggered by alterations in spiking and likely calcium transients associated with cellular spiking; once the cell drifted outside the set point for spiking a cell-wide signal was activated that changed the synaptic strengths of all AMPAergic inputs by a single multiplicative scaling factor to return the cell to the spiking set point (*Turrigiano, 2012*). In this way, AMPAergic scaling could homeostatically regulate spiking levels, while also preserving the relative differences in synaptic strength that have been set up by Hebbian plasticity mechanisms. However, as described in the Introduction, more recent studies suggest that AMPAergic synaptic scaling does not appear to meet these initial expectations. Previous work suggests AMPAergic scaling following TTX or TTX + APV treatment was not multiplicative (*Hanes et al., 2020*; *Wang et al., 2019*), and we now show that it is not multiplicative following AMPAR blockade (CNQX treatment, *Figure 2—figure supplement 1*). Further, several studies suggest that changes in mEPSC amplitude associated with AMPAergic scaling occur at the level of the synapse rather than globally throughout the cell. In fact, several studies have suggested that glutamate receptor activation due to action potential-independent spontaneous release could play a significant role in triggering AMPAergic scaling (*Sutton et al., 2006*; *Fong et al., 2015*; *Aoto et al., 2008*). It is certainly possible that there are compensatory changes in mEPSC amplitude that can be triggered by either altered neurotransmission or spiking. However, as we have shown previously, putting back significant spiking activity levels and their associated calcium transients in the presence of CNQX had no effect on

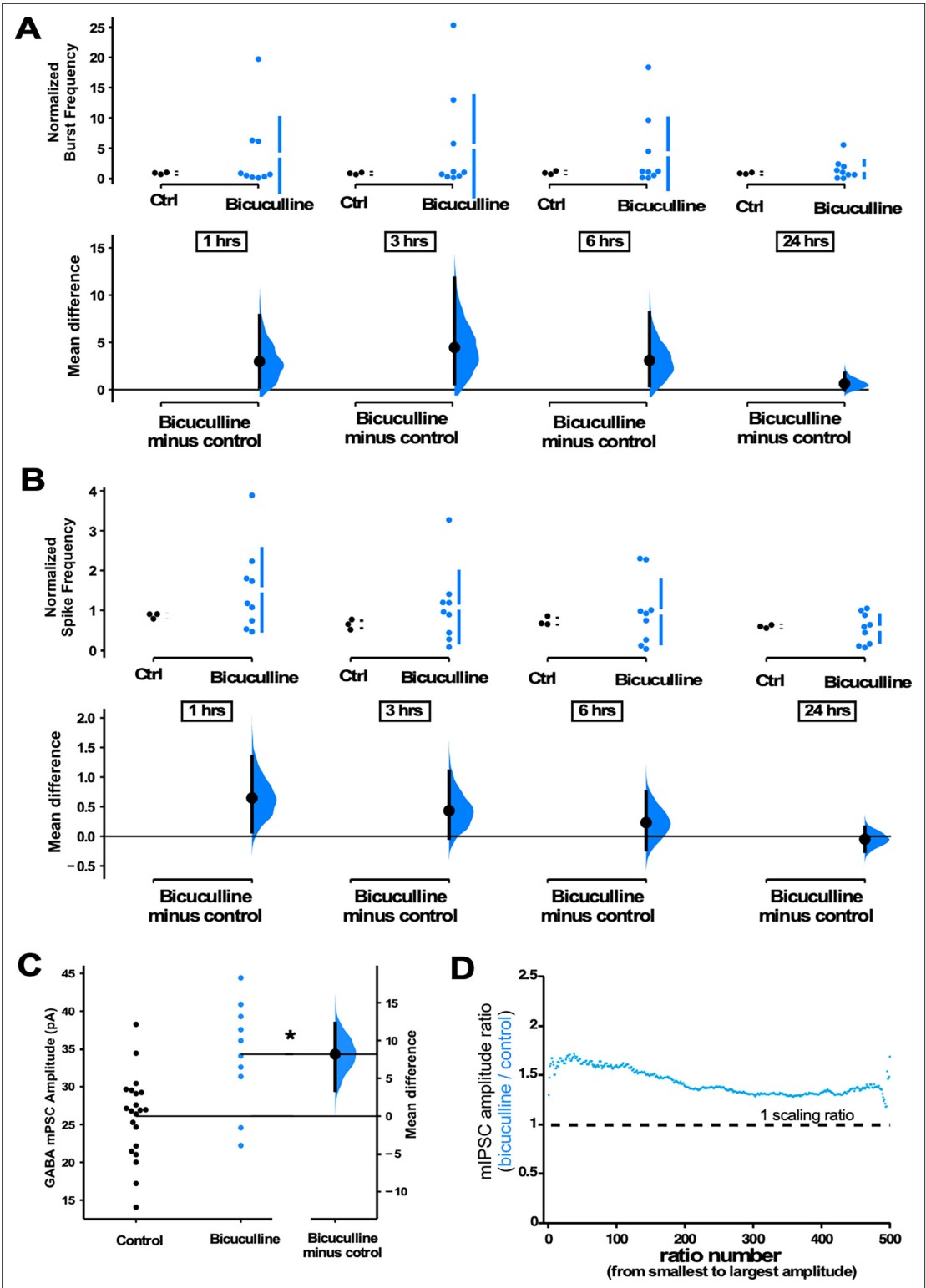

**Figure 6.** GABAergic upscaling is triggered by increased spiking activity rather than reduced GABAR activation. (**A**) Bicuculline-treated cultures (24 hr) plated on multi-electrode arrays (MEAs) trended upward in normalized burst rate compared to control untreated cultures at 1 hr (p=0.63), 3 hr (p=0.556), 6 hr (p=0.547), and 24 hr (p=0.559) after addition of bicuculline (n=9 cultures) or vehicle (n=3 cultures, same data as *Figure 1*). (**B**) Bicuculline-treated cultures (24 hr) plated on MEAs trended upward in normalized overall spike frequency compared to control untreated cultures at 1 hr (p=0.358), 3 hr (p=0.462), 6 hr (p=0.734), and 24 hr (p=0.772) after addition of bicuculline or vehicle. Recordings from single cultures (filled circles), where mean values (represented by the gap in the vertical bar) and SD (vertical bars) are plotted on the upper panels. (**C**) Bicuculline treatment (24 hr) produced an increase

*Figure 6 continued on next page*

*Figure 6 continued*

in miniature inhibitory postsynaptic current (mIPSC) amplitudes (control - n=21 from 10 cultures, bicuculline - n=10 from 4 cultures). The mean difference is compared to control and displayed in Cumming estimation plots. Significant difference denoted by *p≤0.05. Recordings from single neurons (filled circles), and mean values (represented by the horizontal line). Control and treated group is plotted, as a bootstrap sampling distribution (mean difference is represented by a filled circles and the 95% CI is depicted by vertical error bar). (**D**) Ratio plots for bicuculline-induced increase in mIPSCs exhibit a multiplicative profile. All mIPSC amplitudes recorded from cultures plated on coverslips, not MEAs.

The online version of this article includes the following figure supplement(s) for figure 6:

**Figure supplement 1.** Frequency of miniature inhibitory postsynaptic currents (mIPSCs) was no different across conditions.

AMPAergic scaling (no reduction in the existing scaling; *Fong et al., 2015*). Because AMPAergic scaling does not directly follow spiking activity levels, it does not appear to fulfill the expectations of a homeostat for spiking. Rather, AMPAergic scaling in many cases seems to act to homeostatically maintain the effectiveness of individual synapses.

GABAergic scaling appears to exhibit all the features initially predicted for AMPAergic synaptic scaling. First, GABAergic scaling is multiplicative, meaning the relative strengths of these synapses can be maintained (*Figures 2 and 4–6*). Critically, GABAergic scaling can act as a firing rate homeostat for the following reasons. GABAergic downscaling was triggered by alterations in spike rate, rather than AMPAergic neurotransmission. We found that CNQX-triggered GABAergic downscaling was abolished when we optogenetically restored spiking activity levels (*Figures 3 and 4*), and that increasing spiking with bicuculline or CTZ both triggered GABAergic upscaling (*Figures 5 and 6*). While we cannot rule out a role of AMPAR activation in GABAergic upscaling, we did observe that CTZ-induced upscaling was converted to downscaling in the presence of TTX (*Figure 5C and D*). Further, the findings suggest that altering neurotransmission did not contribute to GABAergic scaling. Increasing AMPAergic transmission with CTZ in the presence of TTX had no impact on downscaling as it was no different than following TTX treatment alone (*Figure 5D*). Also, if GABAR transmission

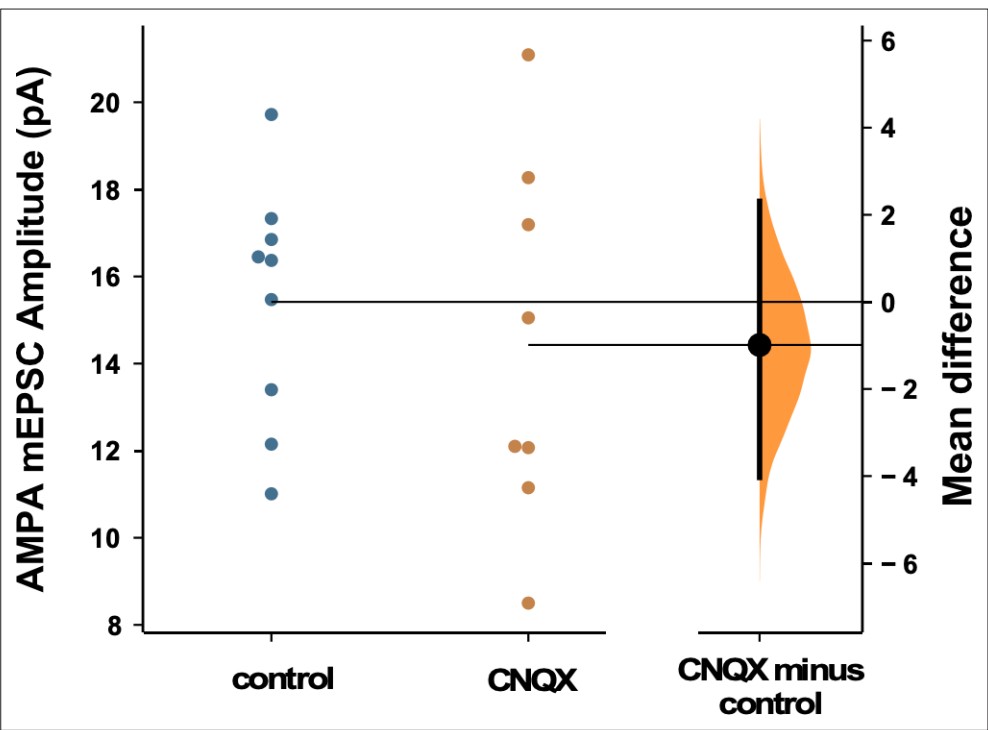

**Figure 7.** AMPAergic scaling was absent following 24 hr of 20 µM CNQX. AMPA mEPSC amplitudes were no different than control following AMPA receptor (AMPAR) blockade (p=0.57, control - n=9 from 4 cultures, CNQX - n=8 from 3 cultures). Recordings from single neurons (filled circles), where mean values (represented by horizontal bar) are plotted, as a bootstrap sampling distribution (mean difference is represented by a filled circles and the 95% CIs are depicted by vertical error bars). All miniature excitatory postsynaptic current (mEPSC) amplitudes recorded from cultures plated on multi-electrode arrays (MEAs).

were a proxy for activity levels, then blocking GABA_A receptors would mimic activity blockade and should lead to a compensatory downscaling. However, bicuculline (reduced GABAR activity) increased spiking and triggered a GABAergic upscaling consistent with the idea that spiking was the critical feature (*Figure 6*). This result was consistent with previous work in hippocampal cultures where chronic bicuculline treatment triggered GABAergic upscaling, which was prevented if the cell was hyperpolarized (*Peng et al., 2010*). Finally, if scaling contributed to a homeostatic recovery of activity, then GABAergic scaling should have been expressed by 24 hr of CNQX (before bursts and before spike frequency in some cultures had fully recovered, *Figure 1*) and this was the case (*Figure 2*). Although AMPAergic scaling was initially thought to play the role of spiking homeostat, it appears more likely that GABAergic scaling is one of the homeostatic mechanisms that is playing this role.

The results of our current study on GABAergic scaling and our previous study on AMPAergic scaling (*Fong et al., 2015*) suggest these two forms of plasticity have distinct triggers and signaling pathways. Optogenetic restoration of activity in CNQX prevented GABAergic downscaling (*Figures 3 and 4*) but had no effect on AMPAergic scaling (*Fong et al., 2015*). Further, increasing glutamatergic receptor activation with CTX during activity blockade reduced TTX-induced AMPAergic scaling (*Fong et al., 2015*) but not GABAergic scaling (*Figure 5D*). We considered the possibility that some of our results could be due to differences in the cultures of this vs our previous study (mouse vs rat, 20 vs 40 µM CNQX, etc.). However, when we reduced spiking activity with 20 µM CNQX and assessed AMPAergic scaling in mouse cortical cultures, we did not trigger AMPAergic scaling at all, again consistent with the idea that the triggers are distinct for these two classes of plasticity. It is not clear to us why we were unable to trigger AMPAergic scaling in this study. It is possible that our cortical cultures (mouse, density) have less capacity for AMPAergic scaling. Alternatively, AMPAergic scaling may require higher concentrations of CNQX to partially influence NMDARs; this could occur through more complete blockade of AMPARs whose depolarization is important in removing the $Mg^{2+}$ block of the NMDAR or through direct block of the glycine binding site of the NMDAR (*Lester et al., 1989*; *Sheardown et al., 1990*). Regardless, the reduction of spiking activity produced by 20 µM CNQX was capable of triggering GABAergic scaling.

Previously, in embryonic motoneurons we found that both GABAergic and AMPAergic scaling was mediated by changes in GABAR activation from spontaneous release rather than changes in spiking activity (*Garcia-Bereguiain et al., 2016*; *Gonzalez-Islas et al., 2018*). However, this was at a developmental stage when GABA was depolarizing and could potentially activate calcium signaling pathways. On the other hand, spike rate homeostasis through the GABAergic system is consistent with many previous studies in which sensory input deprivation *in vivo* led to rapid compensatory disinhibition (*Gainey and Feldman, 2017*; *Ribic, 2020*). For instance, 1 day of visual deprivation (lid suture) reduced evoked spiking in fast spiking parvalbumin (PV) interneurons and this was thought to underlie the recovery of pyramidal cell responses to visual input at this point (*Kuhlman et al., 2013*). One day of whisker deprivation between P17 and P20 produced a reduction of PV interneuron firing that was due to reduced intrinsic excitability in the GABAergic PV neuron (*Gainey et al., 2018*). In addition, 1 day after enucleation, the excitatory to inhibitory synaptic input ratio in pyramidal cells was dramatically increased due to large reductions in GABAergic inputs to the cell (*Barnes et al., 2015*). This disinhibition occurs rapidly (*Hengen et al., 2013*) and can outlast changes in glutamatergic counterparts (*Li et al., 2014*; *Barnes et al., 2015*). These results highlight the important role that inhibitory interneurons play in the homeostatic maintenance of spiking activity. Further, these cells have extensive connectivity with pyramidal cells, placing them in a strong position to influence network excitability (*Fino et al., 2013*; *Packer and Yuste, 2011*). In the current study, we show a critical feature of homeostatic regulation of spiking is through one aspect of inhibitory control, GABAergic synaptic scaling in excitatory neurons.

It is not clear what specific features of spiking trigger GABAergic scaling. GABAergic scaling may require the reduction of spiking in multiple cells in a network, rather than a single cell. Reduced spiking with sporadic expression of a potassium channel in one hippocampal cell in culture did not induce GABAergic scaling in that cell (*Hartman et al., 2006*). Such a result could be mediated by the release of some activity-dependent factor from a collection of neurons. BDNF is known to be released in an activity-dependent manner and has been shown to mediate GABAergic downward scaling following activity block previously in both hippocampal and cortical cultures (*Swanwick et al., 2006*; *Rutherford et al., 1997*). On the other hand, another study increased spiking in hippocampal cultures and

showed that homeostatic increases in mIPSC amplitudes could be dependent on the individual cell's spiking activity (*Peng et al., 2010*). Future work will be necessary to determine the exact feature of spiking that may be more critical in triggering GABAergic scaling (e.g. bursting vs total spike frequency) and the downstream signaling pathway (e.g. somatic calcium transients). While we have no direct support of a role for NMDARs, we cannot rule out the possibility that NMDAR activity could contribute to GABAergic scaling. Previous work has shown that NMDAR block can trigger GABAergic downscaling (*Swanwick et al., 2006*) and our activity manipulations would similarly alter NMDAR activation (CNQX would reduce and optogenetic restoration would restore some NMDAR activation). Whatever the specific features of spiking activity that trigger GABAergic scaling, our results strongly point to the idea that GABAergic scaling could serve a critical role of a spiking homeostat, and highlights the fundamentally important homeostatic nature of GABAergic neurons.

Finally, it is important to take into consideration some of the benefits and limitations of this study. By recording activity levels of cultured neurons through MEAs, we were able to identify the actual influence of the drugs on population activity. This is a step beyond what many homeostatic studies, including our own, typically do, and it affords us the opportunity to interpret more intelligently the results of our perturbations. Regardless, there are limitations associated with these techniques. Cultured networks lack the actual circuitry of the *in vivo* cortex, and for several reasons are vulnerable to dramatic variability (based on plating density, ages, composition, etc.). This variability can be seen in response to drug application throughout our results and it is important to keep in mind that the recorded spiking activity represents the population response from many different classes of excitatory and inhibitory neurons, although the majority are thought to represent excitatory principal neurons. Despite these caveats, the culture system has allowed us to manipulate spiking activity in important ways, which has provided us the insight that GABAergic scaling is one of the homeostatic mechanisms that fulfills the expectations of a spike rate homeostat.

## Materials and methods

### Cell culture

Brain cortices were obtained from C57BL/6J embryonic day 18 mice from BrainBits or harvested from late embryonic cortices. Neurons were obtained after cortical tissue was enzymatically dissociated with papain. Cell suspension was diluted to 2500 live cells per µl and 35,000 cells were plated on glass coverslips or planar MEA coated with polylysine (Sigma, P-3143) and laminin. The cultures were maintained in Neurobasal medium supplemented with 2% B27 and 2 mM GlutaMax. No antibiotics or antimycotics were used. Medium was changed completely after 1 DIV and half of the volume was then changed every 7 days. Spiking activity was monitored starting ~10 DIV to determine if a bursting phenotype was expressed and continuous recordings were made between 14 and 20 DIV. Cultures were discarded after 20 DIV. All protocols followed the National Research Council's Guide on regulations for the Care and Use of Laboratory Animals and from the Animal Use and Care Committee from Emory University.

### Whole-cell recordings

Pyramidal-like cells were targeted based on their large size. Whole-cell voltage clamp recordings of GABA mPSCs were obtained using an AxoPatch 200B amplifier, controlled by pClamp 10.1 software, low pass filtered at 5 kHz online and digitized at 20 kHz. Tight seals (>2 GΩ) were obtained using thin-walled borosilicate glass microelectrodes pulled to obtain resistances between 7 and 10 MΩ. The intracellular patch solution contained the following (in mM): CsCl 120, NaCl 5, HEPES 10, $MgSO_4$ 2, $CaCl_2$ 0.1, EGTA 0.5, ATP 3, and GTP 1.5. The pH was adjusted to 7.4 with KOH. Osmolarity of patch solution was between 280 and 300 mOsm. Artificial cerebral-spinal fluid (ACSF) recording solution contained the following (in mM): NaCl 126, KCl 3, $NaH_2PO_4$ 1, $CaCl_2$ 2, $MgCl_2$ 1, HEPES 10, and D-glucose 25. The pH was adjusted to 7.4 with NaOH. GABAergic mPSCs were isolated by adding to ACSF (in µM): TTX 1, CNQX 20, and APV 50. AMPAergic mPSCs were isolated by adding to ACSF (in µM): TTX 1, APV 50, and gabazine 5. Membrane potential was held at –70 mV and recordings were performed at room temperature. Series resistance during recordings varied from 15 to 20 MΩ and were not compensated. Recordings were terminated whenever significant increases in series resistance (>20%) occurred. Analysis of GABA mPSCs was performed blind to condition with MiniAnalysis

software (Synaptosoft) using a threshold of 5 pA for mPSC amplitude (50 mPSCs were taken from each cell and their amplitudes were averaged and each dot in the scatter plots represents the average of a single cell). Ratio plots of mIPSCs were constructed by taking a constant total number of mIPSCs from control and drug-treated cultures (e.g. 15 control cells with 40 mIPSCs from each cell and 20 CNQX-treated cells with 30 mIPSCs from each cell, 600 mIPSCs per condition). Then the amplitudes of mIPSCs from each condition were rank ordered from smallest to largest and plotted as a ratio of the drug-treated amplitude divided by the control amplitude, as we have described previously (*Hanes et al., 2020*; *Koesters et al., 2024*; *Pekala and Wenner, 2022*).

## MEA recordings

Extracellular spiking was recorded from cultures plated on planar 64-channel MEAs (Multichannel Systems) recorded between 14 and 20 DIV in Neurobasal media with B27 and GlutaMax, as described above. Cultured MEAs were covered with custom-made MEA rings with gas-permeable ethylene-propylene membranes (ALA Scientific Instruments). Synapse software (Tucker-Davis Technologies TDT) was used to monitor activity on a TDT electrophysiological platform consisting of the MEA MZ60 headstage, the PZ2 pre-amplifier, and a RZ2 BioAmp Processor. Recordings were band-pass filtered between 200 and 3000 Hz and acquired at 25 kHz. MEAs were placed in the MZ60 headstage, which was housed in a 5% $CO_2$ incubator at 37°C. Drugs were added separately in a sterile hood and then returned to the MEA recording system. MEA spiking activity was analyzed offline with a custom-made MATLAB program. The recordings acquired in Synapse software (TDT) were subsequently converted using the subroutine TDT2MAT (TDT) to MATLAB files (Mathworks). The custom-written MATLAB program identified bursts of network spikes using an interspike interval-threshold detection algorithm (*Bakkum et al., 2013*). Spiking activity was labeled as a network burst when it met a user-defined minimum number of spikes (typically 10) occurring across a user-defined minimum number of channels (5–10) within a Time-Window (typically 0.1–0.3 s) selected based on the distribution of interspike intervals (typically between the first and tenth consecutive spike throughout the recording, *Figure 1—figure supplement 1*). This program allowed us to remove silent channels and channels that exhibited high-noise levels. The identified network bursts were then visually inspected to ensure that these parameters accurately identified bursts. The program also computed network burst metrics including burst frequency, overall spike frequency, and other characteristics.

## Optogenetic control of spiking

For optostimulation experiments neurons were plated on 64-channel planar MEAs and transfected with AAV9-hSynapsin-ChR2(H134R)-eYFP (ChR2) produced by the Emory University Viral Vector Core. All cultures used in ChR2 experiments, including controls, were transfected at 1 DIV. The genomic titer was $1.8 \times 10^{13}$ vg/ml. Virus was diluted 1–10,000 in growth medium and this dilution was used for the first medium exchange at DIV 1. Finally, the media containing the virus was washed out after 24 hr incubation. A 3 hr pre-drug recording was obtained in the TDT program that determined the average MEA-wide firing rate before adding CNQX. This custom-written program from TDT then delivered a TTL pulse (50–100 ms) that drove a blue light photodiode (465 nm, with a range from 0 to 29.4 mW/mm$^2$, driven by a voltage command of 0–4 V) from a custom-made control box that allowed for scaled illumination. The photodiode sat directly below the MEA for activation of the ChR2. This triggered a barrage of spikes resulting in a burst that looked very similar to a naturally occurring burst not in the presence of CNQX. The program measured the MEA-wide spike rate every 10 s and if the rate fell below the set value established from the pre-drug average, an optical stimulation (50–100 ms) was delivered triggering a burst which then increased the average firing rate, typically above the set point.

## Statistics

Estimation statistics have been used throughout the manuscript. 5000 bootstrap samples were taken; the confidence interval is bias-corrected and accelerated. The p value(s) reported are the likelihood(s) of observing the effect size(s), if the null hypothesis of zero difference is true. For each permutation p value, 5000 reshuffles of the control and test labels were performed (moving beyond p values: data analysis with estimation graphics; *Ho et al., 2019*).

## Acknowledgements

We would like to thank Bill Goolsby who custom built our optogenetic stimulator, and Tucker Davis Technologies for helping us write the Synapse Program that ran the MEA recording/optogenetic stimulation software. We would also like to thank Dr. Gary Bassell for providing us with some of the mice used in culture experiments.

## Additional information

### Funding

| Funder | Grant reference number | Author |
| --- | --- | --- |
| National Institute of Neurological Disorders and Stroke | R01NS065992 | Peter Wenner |
| National Institute of Neurological Disorders and Stroke | R21NS084358 | Peter Wenner |

The funders had no role in study design, data collection and interpretation, or the decision to submit the work for publication.

### Author contributions

Carlos Gonzalez-Islas, Conceptualization, Data curation, Formal analysis, Supervision, Investigation, Methodology, Writing – original draft, Writing – review and editing; Zahraa Sabra, Resources, Software, Formal analysis, Methodology, Writing – review and editing; Ming-fai Fong, Resources, Software, Formal analysis, Investigation, Methodology, Writing – review and editing; Pernille Yilmam, Resources, Methodology, Writing – review and editing; Nicholas Au Yong, Kathrin Engisch, Writing – review and editing; Peter Wenner, Conceptualization, Resources, Data curation, Software, Formal analysis, Supervision, Funding acquisition, Validation, Investigation, Visualization, Methodology, Writing – original draft, Project administration, Writing – review and editing

### Author ORCIDs

Carlos Gonzalez-Islas http://orcid.org/0000-0002-3785-4494
Ming-fai Fong http://orcid.org/0000-0002-2336-4531
Kathrin Engisch http://orcid.org/0000-0002-1058-5343
Peter Wenner http://orcid.org/0000-0002-7072-2194

### Ethics

All experimental procedures were approved by the Institutional Animal Care and Use Committee (IACUC Protocol # PROTO201700661) at Emory University and conducted in compliance with the National Institutes of Health Office of Laboratory Animal Welfare Policy.

Reviewer #3 (Public review): https://doi.org/10.7554/eLife.87753.3.sa1
Author response https://doi.org/10.7554/eLife.87753.3.sa2

## Additional files

### Supplementary files

• MDAR checklist

### Data availability

Analyzed data values are publicly available at the following sites: AMPAergic mPSC values from previous study (*Figure 2—figure supplement 1*) can be found at potterlab.bme.gatech.edu. All other data and the code for the MATLABMatlab program that detects burst features can be found at https://github.com/pwenner/Wenner-eLIfe-data/ (copy archived at *Wenner, 2024*).

The following previously published datasets were used:

| Author(s) | Year | Dataset title | Dataset URL | Database and Identifier |
|---|---|---|---|---|
| Fong M | 2016 | Fong et al | https://neurodatasharing.bme.gatech.edu/cloud/Fong%20et%20al/ | neurodatasharing, Fong%20et%20al/ |
| Wagenaar DA, Pine J, Potter SM | 2005 | Network activity of developing cortical cultures *in vitro* | https://potterlab.bme.gatech.edu/development-data/html/index.html | Potter Lab, development-data |

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
