## [Editor Report · eLife assessment]

This is an **important** study that brings insight into mechanisms that underlie regulation of GABAergic transmission in response to changes in activity. The authors present **solid** data supporting the premise that action potential firing rather than excitatory synaptic strength is a key determinant of GABAergic synaptic inputs.

---

## [Referee Report · Reviewer #3 (Public review)]

This paper concerns whether synaptic scaling (or homeostatic synaptic plasticity; HSP) occurs similarly at GABA and Glu synapses and comes to the surprising conclusion that these can be regulated independently. In fact, under the conditions used in this study, only the GABAergic synapses show HSP and the glutamatergic synapses don't change. This is surprising because these were thought to be co-regulated during HSP and in fact, the major mechanisms thought to underlie downscaling (TTX or CNQX driven), retinoic acid and TNF, have been shown to regulate both GABARs and AMPARs directly. Thus, the main result, that GABA HSP is dissociable from Glu HSP, is novel and exciting. This suggests either different mechanisms underlie the two processes, or that under certain conditions, another mechanism is engaged that scales one type of synapse and not the other. Given that glutamatergic synapses are unchanged in their conditions, that later seems more likely - a novel form of HSP exists that only scale GABAergic synapses. Whether glutamatergic and GABAergic synapses scale independently during HSP affecting both types of synapses remains to be addressed. It would be necessary to demonstrate the dissociation in the same system, under conditions where both types of synapses are changing. But because the form of HSP studied here appears different than that studied in Fong et al., the authors should be careful when comparing the two results. There seems to be an implicit underlying assumption that there is a simple form of HSP, when the overall literature (and the two studies from this lab) supports the idea of many forms of HSP.

The homeostatic changes at GABAergic synapses do seem to be more consistent in amplitude across the bulk of the synapses, which does suggest that true scaling (a proportional change to all synapses on a cell) is occurring. This may represent a major difference in how homeostatic changes occur at the two types of synapses.

The second finding is that this form of HSP seems more regulated by action potential firing than conventional HSP - previous work from this lab had shown that restoring AP firing during AMPA receptor blockade did not prevent scaling of glutamatergic synapses (it should be noted these experiments were done in rat cultures, not mouse, used a higher concentration of CNQX, and used a different optogenetic stimulation paradigm). Restoring AP firing rates under the conditions used here (and thus the form of HSP only affecting GABA synapses), on the other hand, did prevent the homeostatic response. This suggests that this GABA-only form of HSP is more attuned to spiking rates than other forms.

However, details in the data may suggest that spiking is not the (or the only) homeostat, as TTX and CNQX causes identical changes in mIPSC amplitude but have different effects on spiking (although TTX may be driving a different form of HSP). Further, in Fig 5, CTZ had a minimal effect on spiking but a large effect on mIPSCs. Similar issues appear in Fig 6, where the induction of increased spiking is highly variable, with many cells showing control levels or lower spiking rates. Yet the synaptic changes are robust, across all cells. Overall, more will need to be done to conclude that spiking is the homeostat for GABA synapses.

The paper also suggests that the GABA changes are leading to the recovery of the spiking rates, but while they have the time course of the spiking changes and recovery, they only have the 24h time point for synaptic changes. It is not yet possible to conclude how the time courses align without more data, nor can we assume that cells that did not recover to control firing rates would do so eventually.

---

## [Author Response]

The following is the authors’ response to the original reviews.

**eLife assessment**
This study assesses homeostatic plasticity mechanisms driven by inhibitory GABAergic synapses in cultured cortical neurons. The authors report that up- or down-regulation of GABAergic synaptic strength, rather than excitatory glutamatergic synaptic strength, is critical for homeostatic regulation of neuronal firing rates. The reviewers noted that the findings are potentially important, but they also raised questions. In particular, the evidence supporting the findings is currently incomplete and demonstration of independent regulation of mEPSCs and mIPSCs is a necessary experiment to support the major claims of the study.

We appreciate the detailed, thoughtful assessment of our paper by the reviewers and editors and now submit a revised version that addresses the reviewers’ comments as detailed below in response to each concern. We include a more open discussion of alternative possibilities and have added experiments demonstrating that AMPAergic scaling in our mouse cortical cultures is triggered differently than GABAergic scaling. We treated the cultured neurons exactly as described for triggering GABAergic scaling (20µM CNQX for 24 hours), however this did not trigger AMPAergic upscaling (new Figure 7), even though it did reduce spiking/bursting activity. Below we explain the result further, but ultimately this does demonstrate independent regulation of mEPSCs and mIPSCs as requested by the editor/reviewer (spike reductions induced by CNQX reduced mIPSC amplitude, but had no effect on mEPSC amplitude).

**Reviewer #1 (Public Review):**
While the paper is ambitious in its rhetorical scope and certainly presents intriguing findings, there are several serious concerns that need to be addressed to substantiate the interpretations of the data. For example, the CTZ data do not support the interpretations and conclusions drawn by the authors. Summarily, the authors argue that GABAergic scaling is measuring spiking (at the time scale of the homeostatic response, which they suggest is a key feature of a homeostat) yet their data in figure 5B show more convincingly that CTZ does not influence spiking levels - only one out of four time points is marginally significant (also, I suspect that the bootstrapping method mentioned in line 454-459 was conducted as a pairwise comparison of distributions. There is no mention of multiple comparisons corrections, and I have to assume that the significance at 3h would disappear with correction).

We certainly understand the criticism here (similar to reviewer 2’s third point). We now discuss these complications in a more detailed description in the manuscript (CTZ section of results and at end of the discussion). First, we are presenting our entire dataset to be as transparent as possible. Unlike most synaptic scaling studies (including our own) that apply drugs to alter activity and assess mPSC amplitude at the final time point, here we are actually showing CTZ’s effect on spiking activity within the culture over time. This is critical because it has informed us of the drug’s true effect on spiking, the variability that is associated with these perturbations, and the ability and timing of the cultured network to homeostatically recover initial levels. This was important because it revealed that the drugs do not always influence activity in the way we assume, and this provides greater context to our results. Second, we are showing all of our data, and presenting it using estimation statistics which go beyond the dichotomy of a simple p value yes or no (Ho J, Tumkaya T, Aryal S, Choi H, Claridge-Chang A. 2019. Moving beyond P values: data analysis with estimation graphics. Nat Methods 16: 565-66). Estimation statistics have become a more standard statistical approach in the last 15 years and is the preferred method for the Society for Neuroscience’s eNeuro Journal. This method shows the effect size and the confidence interval of the distribution. For the 3 hr time point in Fig. 5B the CTZ/ethanol vs. ethanol data points exhibit very little overlap and the effect size demonstrates a near doubling of spike frequency, and the confidence interval shows a clear separation from 0. This was a pairwise comparison as we compared values at each time point after the addition of ethanol or ethanol/CTZ. Third, the plots illustrate an upward trend in spike frequency at 1 and 6 hrs, but that there is also clear variability. It is important to note that these are multiunit recordings and not purely excitatory principal neurons that we target for mPSC recordings. This complication along with the variability inherent in these cultures could make simple comparisons difficult to interpret and we now discuss this (end of discussion). Regardless, we do see some increase in spiking with CTZ and we clearly see increases in mIPSC amplitude, thus providing some support for the idea that spiking could be a critical player in terms of GABAergic scaling, particularly when put in the context of all of our findings. Future work will be necessary to determine how alterations in spiking lead to changes in mIPSC amplitude and we now discuss this (2nd to last paragraph in discussion).

Then, the fact that TTX applied on top of CTZ drives an increase in mIPSC amplitude is interpreted as a conclusive demonstration that GABAergic scaling is sensing spiking. It is inevitable, however, that TTX will also severely reduce AMAP-R activation - a very plausible alternative explanation is that the augmentation of AMPAR activation caused by CTZ is not sufficient to overcome the dramatic impact of TTX. All together, these data do not provide substantial evidence for the conclusion drawn by the authors.

We believe that the most parsimonious explanation for our results is that spiking activity, not AMPAR activation, triggers GABAergic downscaling. GABAergic scaling is no different when comparing 24hr TTX treatment vs TTX+CTZ, and optogenetic restoration of spiking activity while continuing to block AMPAR activation was able to restore GABAergic mPSC amplitudes to control levels. It is important to emphasize that our results with TTX vs. TTX+CTZ are different for GABAergic scaling (no difference in this study) and AMPAergic scaling (CTZ diminished upward scaling in previous study – Fong et al., 2015 - PMID: 25751516) suggesting different triggers for the two forms of scaling. While we strongly believe we have demonstrated that GABAergic downscaling is dependent on spiking (not AMPAergic transmission), we now acknowledge that we cannot rule out the possibility that upward GABAergic scaling may be influenced by AMPAR activation (2nd paragraph discussion), although we have no evidence in support of this.

Specific points:- The logic of the basis for the argument is somewhat flawed: A homeostat does not require a multiplicative mechanism, nor does it even need to be synaptic. Membrane excitability is a locus of homeostatic regulation of firing, for example. In addition, synapse-specific modulation can also be homeostatic. The only requirement of the homeostat is that its deployment subserves the stabilization of a biological parameter (e.g., firing rate).

We largely agree with the reviewer and should not have implied that this was a necessary requirement for a spike rate homeostat. What we should have said was that historically this definition has been applied to AMPAergic scaling, which is thought to be a spike rate homeostat. We have now corrected this (introduction and discussion).

- Line 63 parenthetically references an important, but contradictory study as a brief "however". Given the tone of the writing, it would be more balanced to give this study at least a full sentence of exposition.

Agreed, and we have now done this.

- The authors state (line 11) that expression of a hyperpolarizing conductance did not trigger scaling. More recent work ('Homeostatic synaptic scaling establishes the specificity of an associative memory') does this via expression of DREADDs and finds robust scaling.

The purpose of citing this study was to argue that the spike rate homeostat hypothesis doesn’t make sense for AMPAergic scaling based on a study that hyperpolarized an individual cell while leaving the rest of the network unaltered and therefore leaving network activity and neurotransmission largely normal. In this previous study scaling was not triggered, suggesting reduced spike rate within an individual cell was insufficient to trigger scaling in that cell. The more recent study mentioned by the reviewer achieved scaling by hyperpolarizing a majority of cells in the network. Importantly, this approach alters neurotransmission throughout the network, making it challenging to isolate the specific contributions of spiking vs. receptor activation. Unlike the previous study, which focused on the impact within individual cells, this newer study involves global alterations in network activity, complicating the interpretation of the role of spiking versus receptor activation in triggering scaling.

- Supplemental figure 1 looks largely linear to me? Out of curiosity, wouldn't you expect the left end to be aberrant because scaling up should theoretically increase the strength of some synapses that would have been previously below threshold for detection?

We agree that the scaling ratio plot is largely linear. To be clear, the linearity of the ratio plot was not our point here, rather that there was a positive slope meaning ratios (CNQX mEPSC amplitudes/control mEPSC amplitudes) got bigger for the larger CNQX-treated mEPSCs. Alternatively, a multiplicative relationship where mEPSCs are all increased by a single factor (e.g. 2X) would be a flat line with 0 slope at the multiplicative value (e.g. 2). In terms of the left side of the plot, we do see values that rise abruptly from 1 - this was partially obstructed by the Y axis in this figure and we have adjusted this. This left part of the plot is likely due the CNQX-induced increases in mEPSC amplitudes of mini’s that where below our detection threshold of 5pA, as suggested by the reviewer. Therefore, mini’s that were 4pAs could now be 5pAs after CNQX treatment and these are then divided by the smallest control mEPSCs which are 5 pAs (ratio of 1). We tried to do a better job describing this in the resubmission (1st paragraph of results).

- Given that figure 2B also shows warping at the tail ends of similar distributions, how is this to be interpreted?

The left side of the ratio plot shows evidence consistent with the idea that mIPSCs are dropping into the noise after CNQX treatment (smallest GABA mIPSCs that don’t fall into noise are 5pA and this is divided by the smallest control GABA mPSCs of 5pPA and therefore the ratio is 1). The rest of the distribution will then approach the scaling factor (50% in this case). On the right side of the ratio plot the values appear to slightly increase. We are not sure why this is happening, but it maybe that a small percentage of mIPSCs are not purely multiplicative at 0.5, however the biggest mPSCs can vary to a great degree from one cell to the next and in other cases we do not see this (Figure 4B, Figure 5E). We tried to do a better job describing this in the resubmission (results describing Figure 2).

- The readability of the figures is poor. Some of them have inconsistent boundary boxes, bizarre axes, text that appears skewed as if the figures were quickly thrown together and stretched to fit.

We have adjusted the figures to be more consistent throughout the manuscript.

- I'm concerned about the optogenetic restoration of activity experiment. Cortical pyramidal neuron mean firing rates are log normally distributed and span multiple orders of magnitude. The stimulation experiments can only address the total firing at a network-level - given than a network level "mean" is meaningless in a lognormal distribution, how are we to think about the effect of this manipulation when it comes to individual neurons homeostatically stabilizing their own activities? In essence, the argument is made at the single-neuron level, but the experiment is conducted with a network-level resolution.

As described above, we do not have the capacity to know what the actual firing rate of a particular neuron was before and after perturbing the system, and certainly not for the specific cells we recorded from to obtain mPSC amplitudes, and so we cannot say that we have perfectly restored the original firing rates of neurons. However, there is reason to believe that this is achieved to some extent. Our optogenetic stimulation is only 50-100 ms long activating a subset of neurons. This is sufficient to provide a synaptic barrage that then triggers a full blown network burst where the majority of spikes occur, but this is after the light is off. In other words, the optogenetic light pulse only initiates what becomes a relatively normal network burst that fortunately allows the individual cells to express their relatively normal (pre-drug) activity pattern. In our previous study using optogenetic activity restoration (Fong et al., 2015) we were able to show that this was the case for individual units - the spiking of an individual unit during a burst is similar before and after CNQX/optogenetic stimulation (see Figure 4b and Suppl. Fig 4 in Fong et al. 2015). We are not claiming that we have restored spiking to exactly the pre-drug state, but bring it back toward those levels and we see this is associated with a return of the mIPSC amplitude to near control levels. We now include a brief description of this in the manuscript (results describing Figure 3).

- Line 198-99: multiplicativity is not a requirement of a homeostatic mechanism.- Line 264-265 - again, neither multiplicativity and synaptic mechanisms are fundamentally any more necessary for a homeostatic locus than anything else that can modulate firing rate in via negative feedback.

As mentioned above, the multiplicative nature of scaling has been a historical proposal for AMPAergic scaling and we have now found such a relationship for GABAergic scaling. This is important for understanding how this plasticity works, but we agree that it is not necessary for a homeostat and we have adjusted the manuscript accordingly.

- 277: do you mean AMPAR?

We were not clear enough here. We actually do mean GABAR. The idea was that CTZ increases network activity and thus increases both AMPAergic and GABAergic transmission. We have rewritten this part of the discussion to avoid any confusion (2nd paragraph discussion).

- Example: Figure 1A is frustratingly unreadable. The axes on the raster insets are microscopic, the arrows are strangely large, and it seems unnecessary to fill so much realestate with 4 rasters. Only one is necessary to show the concept of a network burst. The effect of time+CNQX on the frequency of burst is shown in B and C.- Example: Figure 2 appears warped and hastily assembled. Statistical indications are shown within and outside of bounding boxes. Axes are not aligned. Labels are not aligned. Font sizes are not equal on equivalent axes.

These figures were generated by the estimation statistics website and text may have been resized inappropriately. We have tried to adjust this and now have attempted to standardize the axes text to the best of our ability.

- The discussion should include mention of the limitations and/or constraints of drawing general conclusions from cell culture.

We have added this consideration at the end of the discussion. Further, this is why we cited studies that argue GABAergic neurons have a particularly important role in homeostatic regulation of firing following sensory deprivations *in vivo*.

- The discussion should include mention of the role of developmental age in the expression of specific mechanisms. It is highly likely that what is studied at ~P14 is specific to early postnatal development.

We now discuss caveats of cortical cultures at the end of the discussion.

It is essential to ensure that the data presented in the paper adequately supports the conclusions drawn. A more cautious approach in interpreting the results may lead to a stronger argument and a more robust understanding of the underlying mechanisms at play.

We have broadened our discussion of alternative interpretations throughout the manuscript.

**Reviewer #1 (Recommendations For The Authors):**
While I am hesitant to judge a paper based on its tone, I would personally recommend revision of some of the subjective words and statements, as the manuscript undermines its own effectiveness by making unnecessarily strong statements. The text repeatedly paints an "either A or B" picture, and if there's any general lesson in biology, it's that it's always A and B. Global, multiplicative glutamatergic scaling could quite conceivably occur alongside GABAergic scaling, as well as synapse-specific homeostatic modifications. It seems that it would be wise to acknowledge that, while the data presented here point in one direction, in vivo results in an adult brain (for example) might present an entirely different set of patterns. This will not only enhance the readability of the paper but also ensure that the scientific community can engage with the work in a constructive and collaborative manner. Again, I present this as only a constructive and supportive suggestion. I am a big fan of work from this laboratory, and I would love to see this paper in an improved form - it's an important set of ideas and I do believe that these data are rigorously collected.

We have attempted to provide a more comprehensive interpretation of our results. We agree that a homeostat can come in many flavors, but do believe that GABAergic scaling is strong candidate, whereas AMPAergic scaling does not currently fit such a role. We do now discuss caveats with our work and are open to other interpretations that need to be flushed out in future work.

**Reviewer #2 (Public Review):**
Major points:(1) The reason why CNQX does not completely eliminate spiking is unclear (Fig. 1). What is the circuit mechanism by which spiking continues, although at lower frequency, in the absence of AMPA-mediated transmission and what the mechanism by which spiking frequency grows back after 24h (still in the absence of AMPA transmission)?Is it possible that NMDA-mediated transmission takes over and triggers a different type of network plasticity?

The bursting in AMPAR blockade is due to the remaining NMDA receptor-mediated transmission. We showed this in our previous study in Suppl. Figure 2 and 6 of Fong et al., 2015 (PMID: 25751516). Our ability to optically induce normal looking bursts of spikes was also dependent NMDAR activation (Fong et al 2015 and Figure 6 Newman et al., 2015 - PMID: 26140329). Further, in Dr Fong’s PhD dissertation it was shown that the bursting activity was abolished when AMPA and NMDA receptors were both blocked. There are likely many factors that contribute to the recovery of activity, and certainly one of them is likely to be the weakening of inhibitory GABAergic currents as we had mentioned. We have now added the point about NMDARs mediating the remaining bursts in the manuscript (results associated with Figure 1). We are not clear on what the reviewer has in mind in terms of “NMDA-mediated transmission takes over and triggers a different kind of network plasticity”, but we do discuss the possibility that spiking triggers GABAergic scaling through its effect on NMDAergic transmission, which we cannot rule out, but also have no evidence in support of this idea (3rd and 5th paragraph of discussion). We do plan on addressing this in a future work.

(2) A possible activation of NMDARs should be considered. One would think that experiments involving chronic glutamatergic blockade could have been conducted in the presence of NMDAR blockers. Why this was not the case?

Unfortunately, it was not possible to optogenetically restore normal bursting in the presence of NMDAR blockade (even when AMPAergic transmission was intact), as NMDARs appeared to be critical for the optical restoration of the normal duration and form of the burst in rat cortical cultures (see Suppl. Figure 6 Fong et al., 2015 Nat Comm and Figure 6 Newman et al., 2015). Even high concentrations of CNQX (40µM) prevented us from restoring spiking in mouse cultures in the current study, which is why we moved to 20µM CNQX for this study. The reviewer raises an excellent point about a possible NMDAR contribution to altered synaptic strength, however. It is likely that NMDAR signaling is reduced in the presence of CNQX since burst frequency was dramatically reduced along with AMPAR-mediated depolarizations. We cannot rule out the possibility that NMDAR signaling could contribute to the alterations in GABAergic mIPSCs and discuss this in the resubmission (3rd and 5th paragraph of the discussion). We had not considered this previously because prior work suggested that 24/48 hour block NMDARs (APV) did not trigger AMPAergic scaling in cortical or hippocampal cultures (see Figure 1 Turrigiano et al., 1998 Nature and Suppl. Figure 4 Sutton et al., 2006 Cell), moreover, our previous study showed that restoring NMDAergic transmission ontogenetically, at least to some extent, had no influence on AMPAergic scaling (Fong et al., 2015).

Also, experiments with global ChR2 stimulation with coincident pre and postsynaptic firing might also activate NMDARs and result in additional effects that should be taken into consideration for the global scaling mechanism.

To be clear, our optical stimulation was of short duration (duration 50-100 ms) and was turned off before the vast majority of spiking that occurred in the bursts. So the light flash was a trigger that allowed a relatively normal looking burst to occur after the light was off (see lower panel of Figure 3B optogenetic stimulation – short duration only at onset of burst – we now make this clearer in resubmission). Therefore, we were unlikely to trigger significant synchronous activation that does not normally occur in network bursts.

(3) Cultures exposed to CTZ to enhance AMPA receptors generated variable results (Fig. 5), somewhat increasing spiking activity in a non-significant manner but, at the same time, strengthening mIPSC amplitude. This result seems to suggest that spiking might be involved in GABAergic scaling, but it does not seem to prove it. Then, addition of TTX that blocked spiking reduced mIPSC amplitude. It was concluded here that the ability of CTZ to enhance GABAergic currents was primarily due to spiking, rather than the increase in AMPA-mediated currents. However, in addition to blocking action potentials, TTX would also prevent activation of AMPARs in the presence of CTZ due to the lack of glutamatergic release. Therefore, under these conditions, an effect of glutamatergic activation on GABAergic scaling cannot be ruled out.

These concerns were very similar to reviewer 1’s first comments (see above). To be clear we are going a step beyond most scaling studies by assessing MEA-wide firing rate, but this still provides an incomplete picture of the particular cells that we target for patch recordings in terms of their firing before and after a drug. Further, we see considerable variability in effect on firing rate from culture to culture, which we now discuss in the resubmission (final paragraph discussion). The fact that mIPSCs are no different after TTX treatment vs CTZ+TTX treatment suggests that AMPAergic transmission is not so influential on GABAergic downscaling. While the CTZ results are not conclusive by themselves, taken together with the optogenetic results, where restoration of spiking in AMPAR blockade reverses scaling, is most consistent with idea that GABAergic scaling is triggered by spiking rather than AMPAR activation and places GABAergic scaling as a strong candidate as spike rate homeostat. Although we do feel that we have demonstrated that downward GABAergic scaling is dependent on spiking, we cannot rule out the possibility that upward GABAergic scaling could be influenced by AMPAR activation to some extent. We now acknowledge this possibility (2nd paragraph discussion).

(4) The sample size is not mentioned in any figure. How many cells/culture dishes were used in each condition?

The individual dots represent either individual cells for mIPSC amplitude or individual cultures in MEA experiments. Number of cultures and cells are now stated in the figure legends.

(5) Cortical cultures may typically contain about 5-10% GABAergic interneurons and 90-95 % pyramidal cells. One would think that scaling mechanisms occurring in pyramidal cells and interneurons could be distinct, with different impact on the network. Although for whole-cell recordings the authors selected pyramidal looking cells, which might bias recordings towards excitatory neurons, naked eye selection of recording cells is quite difficult in primary cultures. Some of the variability in mIPSC amplitude values (Fig. 2A for example) might be attributed to the cell type? One could use cultures where interneurons are fluorescently labeled to obtain an accurate representation. The issue of the possible differential effects of scaling in pyramidal cells vs. interneurons and the consequences in the network should be discussed.

We now include this discussion in the resubmission (final paragraph discussion). Briefly, we chose large cells, which will be predominantly glutamatergic neurons as suggested by the reviewer. Ultimately, even among glutamatergic principal cells there may be variability in the response to drug application. All of these issues could contribute to variability and we have expanded our description of the variability in our results, including that based on cellular heterogeneity.

**Reviewer #2 (Recommendations For The Authors):**
Minor comments –Fig S3: Please quantify changes in frequency

We have done this (Supplemental Figure 5).

Fig 2: please choose colors with higher contrast for CNQX/TTX

We have done this.

Fig. 3C: Why doesn't CNQX+PhotoStim reach control levels of bursting at 2h?

The program was designed to follow and maintain total spike frequency and so it does a better job at this than maintaining burst frequency.

Fig. 5A: please include a comparison between control and Ethanol

We now do this in Figure 5C. Both around 26pAs.

Fig. 5C: where is the Etoh condition?

We have made this figure more clear in terms of controls (Figure 5C & D).

**Reviewer #3 (Public Review):**
This paper concerns whether scaling (or homeostatic synaptic plasticity; HSP) occurs similarly at GABA and Glu synapses and comes to the surprising conclusion that these are regulated separately. This is surprising because these were thought to be co-regulated during HSP and in fact, the major mechanisms thought to underlie downscaling (TTX or CNQX driven), retinoic acid and TNF, have been shown to regulate both GABARs and AMPARs directly. (As a side note, it is unclear that the manipulations used in Josesph and Turrigiano represent HSP, and so might not be relevant). Thus the main result, that GABA HSP is dissociable from Glu HSP, is novel and exciting. This suggests either different mechanisms underlie the two processes, or that under certain conditions, another mechanism is engaged that scales one type of synapse and not the other.However, strong claims require strong evidence, and the results presented here only address GABA HSP, relying on previous work from this lab on Glu HSP (Fong, et al., 2015). But the previous experiments were done in rat cultures, while these experiments are done in mice and at somewhat different ages (DIV). Even identical culture systems can drift over time (possibly due to changes in the components of B27 or other media and supplements). Therefore it is necessary to demonstrate in the same system the dissociation. To be convincing, they need to show the mEPSCs for Fig 4, clearly showing the dissociation. Doing the same for Fig 5 would be great, but I think Fig 4 is the key.

We understand the concern of the reviewer as we do see significant variability within our cultures and they were plated in different places, by different people, in different species (rat vs mouse). Therefore, we have attempted to redo the study on AMPAergic scaling on these mouse cortical neurons. Surprisingly, we found that 20µM CNQX did not trigger AMPAergic upscaling (new Figure 7), even though it did reduce spiking activity and was able to produce GABAergic downscaling. We did not carry out the optogenetic restoration of activity, because we did not trigger upscaling. The result does however, show that the reductions in spiking/bursting that trigger GABAergic downscaling, did not trigger AMPAergic upscaling and therefore dissociate the 2 forms of scaling in these mouse cultures. We do not know why 20 µM CNQX did not trigger scaling in these cultures since it does reduce spiking and AMPAR activation. In the Fong study we used 40µM CNQX because intracellular recordings from rat cortical neurons suggested this was required to completely block AMPAergic currents. Our initial studies in the current manuscript examining GABAergic scaling in mouse cortical cultures used 40µM CNQX, however, this concentration of CNQX prevented us from restoring spiking through optogenetic activation, so we reduced our concentration to 20µM CNQX, which did trigger GABAergic downscaling and allowed the restoration of spiking. We now show and discuss this result (Figure 7 and 3rd paragraph discussion).

The paper also suggests that only receptor function or spiking could control HSP, and therefore if it is not receptor function then it must be spiking. This seems like a false dichotomy; there are of course other options. Details in the data may suggest that spiking is not the (or the only) homeostat, as TTX and CNQX causes identical changes in mIPSC amplitude but have different effects on spiking. Further, in Fig 5, CTZ had a minimal effect on spiking but a large effect on mIPSCs. Similar issues appear in Fig 6, where the induction of increased spiking is highly variable, with many cells showing control levels or lower spiking rates. Yet the synaptic changes are robust, across all cells. Overall, this is not persuasive that spiking is necessarily the homeostat for GABA synapses.

Together our results argue against AMPAR or GABAR activation as a trigger for GABAergic scaling and that this is different than our results for AMPAergic scaling. These points alone are important to recognize. While changes in spiking do not perfectly follow the changes in GABAergic scaling they do always trend in the right direction. As mentioned above, total spiking activity is only one measure of spiking. It is possible that these drugs alter the pattern of spiking that translates into an altered calcium transients which may be important for triggering the plasticity. Further, we acknowledge that we cannot rule out a role for NMDARs contributing to GABAergic scaling (3rd and 5th paragraph of discussion). Based on the variability that we observe and the nature of our MEA recordings we cannot precisely determine how the total activity or pattern of activity changes with drug application in the specific cells that we target for whole cell recordings, and this is now discussed (final paragraph of discussion). Again, it is important to note that we are going a step beyond most homeostatic plasticity studies that add a drug and simply assume it is having an effect on spiking (e.g. CNQX was initially thought to completely abolish spiking, but clearly does not). However, we believe that the most parsimonious explanation of our results supports our proposal that GABAergic scaling is a strong candidate as a spike rate homeostat. Regardless, in the resubmission we have included a broader discussion about these possibilities, and recognize that we cannot rule out the possibility that AMPAergic transmission could contribute to upward GABAergic scaling (2nd paragraph discussion).

The paper also suggests that the timing of the GABA changes coincides with the spiking changes, but while they have the time course of the spiking changes and recovery, they only have the 24h time point for synaptic changes. It is impossible to conclude how the time courses align without more data.

We can only say that by the 24 hour CNQX time point, when overall spiking is recovered in some but not all cultures and bursts have not recovered, that GABAergic scaling has already occurred. We now state this more clearly in the resubmission (near the end of the 2nd paragraph of the discussion).

**Reviewer #3 (Recommendations For The Authors):**
The statistics are inadequately described. The full information including actual p values should be given, particularly for the non-significant trends reported.

We have done this in Figure legends.

The abstract and introduction give the impression that GABA and Glu HSP are independent, though most work links them as occurring simultaneously and in a coordinated fashion to achieve homeostasis.

While it is true that many studies have triggered both forms of scaling with activity or transmission blockade, these studies have not addressed whether these forms of scaling are actually triggered in the same way mechanistically, except potentially for the one study that we mentioned (Joseph et al.,). Our results suggest they are independent. We now do mention the idea that these two forms of scaling have been assumed to be commonly triggered (3rd paragraph introduction).

The data in Fig 6 is presented as if BIC treatment is a novel result, although BIC/Gabazine/PTX have been used to induce down-scaling in many previous papers. While it's good to have the results, they should be put in proper context. As suggested in the paper, testing if decreased GABAR function would lead to upscaling does not make sense given all the previous data.

Figure 6 shows GABAergic upscaling in response to GABAR block (bicuculline), but we are aware of only two other studies that looked at GABAergic scaling after treating with a GABAR blocker and they found upscaling but this was in hippocampal cultures, not cortical cultures (Peng et al., 2010 - PMID: 21123568, Pribiag et al., 2014 - PMID: 24753587). We now mention this in the results section describing Figure 6. While many studies have blocked GABARs and find AMPAergic downscaling, we are addressing the triggers for GABAergic scaling in Figure 6.

Is Fig S4B mislabeled? The title says spike rate but the graph axis says burst frequency.

The reviewer is correct and we have now adjusted this.